# TACTIC: Task-Aware Sparse Coordination Graphs for Multi-Task Multi-agent Reinforcement Learning

**Kexing Peng** [1]   **Pengyi Li** [2]   **Tinghuai Ma** [3 1]   **Jianye Hao** [2]

## Abstract

Value factorization eases non-stationarity in MARL, but its static coordination assumptions hinder generalization on long-horizon tasks with shifting dependencies. Prior VQ-VAE methods abstract trajectories yet miss time-varying inter-agent dependencies. We present TACTIC, a CTDE framework with three components: (i) VQ-VAE-based trajectory abstraction that learns discrete task-semantic classes; (ii) semantic-conditioned sparse coordination graphs that adapt dependencies by pruning edges according to variance-based pairwise payoff sensitivity; and (iii) a pretrained, frozen trajectory-class predictor that conditions local policies while decoupling task recognition from control. On SMAC and SUMO, TACTIC shows strong overall competitiveness and adaptive coordination under sparse rewards and dynamic task structures.

## 1. Introduction

In cooperative Multi-Agent Reinforcement Learning (MARL) (Li et al., 2024), value decomposition methods such as QMIX (Rashid et al., 2018) and VDN (Sunehag et al., 2018) address credit assignment and improve scalability by factorizing the joint action-value into agent-level utilities; with recurrent encoders they can operate under partial observability. However, in complex environments with sparse and delayed rewards (Liu et al., 2025), agents must coordinate over long horizons under uncertainty, where vanilla value-decomposition often suffers from inefficient exploration and limited modeling of adaptive, task-aware

coordination. To address this challenge, hierarchical frameworks (Zeng et al., 2025), skill abstractions, and role-based decompositions (Wang et al., 2021b) reduce the search space and encourage committed coordination by conditioning policies on skills/roles/subgoals. Yet many of these approaches implicitly assume relatively stationary task semantics or pre-specified role taxonomies; when the task distribution shifts (e.g., across maps or unit-mix variations as in SMACv2), their generalization tends to degrade.

To address variability and distributional shifts across tasks, recent studies explore multi-task adaptation within a single CTDE training run, where agents learn versatile policies that induce distinct coordination patterns across problems. A promising direction is trajectory-level abstraction, as in LAGMA (Na & Moon, 2024) and TRAMA (Na et al., 2025), which use Vector-Quantized Variational Autoencoders (VQ-VAE) (Li & Yuan, 2021) to embed states or trajectories into quantized latent codes. By clustering semantically similar trajectories, these methods promote efficient exploration and knowledge sharing across related tasks. However, while trajectory-level abstraction facilitates task-aware generalization, it remains largely agnostic to how agents should dynamically coordinate within each abstracted regime (Chen et al., 2025). In contrast, graph-based coordination methods, including DICG (Li et al., 2021), CASEC (Wang et al., 2022), and DMCG (Gupta et al., 2026), adapt inter-agent dependencies to context or graph structure, yet do not explicitly couple such structural adaptation with trajectory-level task semantics.

To address diverse subtask adaptation and dynamic inter-agent dependencies, we propose TACTIC, an integrated CTDE framework that couples VQ-VAE-based trajectory abstraction with a sparse coordination-graph module. TACTIC bridges trajectory semantics and structural coordination by mapping discretized trajectory abstractions into edge-level graph modulation, enabling semantic-to-structural alignment across tasks. To capture evolving coordination patterns, a sparse, context-adaptive coordination graph inspired by DMCG (Gupta et al., 2026) selectively retains edges based on the variance-based pairwise edge relevance (Wu et al., 2024), producing a lightweight yet adaptive topology that reflects the contextual relevance of interactions. Mean-

---

[1]School of Computer Science, School of Software, Nanjing University of Information Science & Technology, Jiangsu, Nanjing, China [2]College of Intelligence and Computing, Tianjin University, Tianjin, China [3]School of Computer Engineering, Jiangsu Ocean University, Jiangsu, Lianyungang, China. Correspondence to: Tinghuai Ma <thma@nuist.edu.cn>.

*Proceedings of the 43rd International Conference on Machine Learning*, Seoul, South Korea. PMLR 306, 2026. Copyright 2026 by the author(s).

while, TACTIC converts discrete trajectory classes obtained from the VQ-VAE into graph-level coordination signals, where each class conditions inter-agent interaction patterns and modulates adaptive coordination-graph edge formation through gated adjacency attention. A pretrained, frozen goal predictor decouples task recognition from control, preventing task-prediction gradients from interfering with policy learning. Through this coherent design, TACTIC couples structural adaptability with semantic consistency, improving generalization across diverse subtasks when latent trajectory semantics remain aligned with coordination structure, while reducing the need for task-specific retraining under shifting task distributions.

To evaluate the effectiveness of TACTIC, we conduct extensive experiments on diverse multi-agent benchmarks, including SMAC (StarCraft Multi-Agent Challenge) (Vinyals et al., 2019) and the urban traffic simulator SUMO (Lopez et al., 2018). On SMACv1 and SMACv2 (Ellis et al., 2023), which involve cooperative control under partial observability and changing unit compositions, TACTIC achieves competitive overall win rates against representative baselines. It performs particularly well in SMACv2 scenarios with sparse rewards and dynamic dependencies, suggesting improved generalization across heterogeneous agent interactions. In SUMO environments, TACTIC improves traffic-efficiency metrics and provides deployment-oriented evidence of coordinated control. Ablation studies further show that task-adaptive coordination graphs, trajectory-class-conditioned policy learning, and trajectory-based latent representations each contribute substantially to overall performance.

We briefly summarize our contributions:

• We introduce a dynamic coordination graph in which task-conditioned VQ-VAE trajectory embeddings modulate edge weights over time, while an action-marginal variance statistic of pairwise payoff functions is used as a practical sensitivity proxy for pruning redundant or weakly relevant interactions. By coupling semantic trajectory abstraction with structural refinement while keeping their optimization signals separated, this design preserves task-relevant connections and enables efficient, context-adaptive coordination with modest computational overhead.

• We design a trajectory-class-aware policy that includes a goal predictor pre-trained and then frozen, preventing gradient interference between task recognition and control learning. By conditioning each agent's decision on stable predicted task classes, the policy remains more consistent across diverse multi-task environments.

• We evaluate TACTIC across diverse domains and tasks, including SMACv1/v2 and SUMO, encompassing reward sparsity, dynamic agent dependencies, and communication delays. TACTIC shows strong overall competitiveness against representative baselines and exhibits favorable cross-task generalization under controlled ablation and stress-testing protocols.

## 2. Background

### 2.1. Dec-POMDPs and Factorized $Q$-Functions

**Dec-POMDPs Setup.** We focus on fully cooperative multi-agent tasks that can be modeled as a Dec-POMDP $\langle I, S, A, P, R, \Omega, O, n, \gamma \rangle$. Here, $I$ is a finite set of $n$ agents, $\gamma \in [0, 1)$ is the discount factor, and $s \in S$ is the true global state. At each timestep $t$, each agent $i \in I$ receives an observation $o_i \in \Omega_i$ from the observation function $O(s, i)$ and chooses an action $a_i \in A_i$. The joint action $\boldsymbol{a} = \langle a_1, \ldots, a_n \rangle$ transitions the environment to a new state $s'$ according to the transition function $P(s' \mid s, \boldsymbol{a})$, yielding a shared reward $r = R(s, \boldsymbol{a}, s')$. Each agent maintains a local action-observation history $\tau_i \in \mathrm{T} \equiv (\Omega \times A)^* \times \Omega$, from which it learns a local policy $\pi_i(a \mid \tau_i)$. We adopt the joint action-value function $Q_{tot}$, defined as $Q_{tot}(s, \boldsymbol{a}) = \mathbb{E}[\sum_{t=0}^{\infty} \gamma^t R(s_t, \boldsymbol{a}_t) \mid s_0 = s, \boldsymbol{a}_0 = \boldsymbol{a}]$, and train policies to maximize the expected return under this Dec-POMDP.

**Coordination Graphs.** To avoid enumerating the full joint action space, a coordination graph $\mathcal{G} = \langle \mathcal{V}, \mathcal{E} \rangle$ factors the joint $Q$-function into local utilities and pairwise payoffs (Kok & Vlassis, 2006). For pairwise edges, one typical factorization is:

$$Q_{\mathrm{tot}}(\boldsymbol{\tau}, \boldsymbol{a}) = \frac{1}{|\mathcal{V}|} \sum_i q_i(\tau_i, a_i) + \frac{1}{|\mathcal{E}|} \sum_{\{i,j\} \in \mathcal{E}} q_{ij}(\tau_i, \tau_j, a_i, a_j).$$

(1)

Under partial observability, we write $Q_{tot}(\boldsymbol{\tau}, \boldsymbol{a})$ with joint histories $\boldsymbol{\tau} = \langle \tau_1, \ldots, \tau_n \rangle$. Most prior works assume $\mathcal{G}$ to be static, while our approach incorporates *dynamic, task-aware* coordination structures (details in Appendix B.1).

**Max-Sum Message Passing.** To obtain coordination-aware utility signals on the task-adaptive coordination graph $\mathcal{G}_t$, we apply Max-Sum message passing on a bipartite factor graph. Agent nodes correspond to actions, while function nodes represent local utility or payoff functions. The edge set $\mathcal{E}_t$ is constructed from VQ-VAE-based trajectory embeddings $z_t$ and updated periodically during training, enabling task-aware graph adaptation. Messages are updated as follows:

*Agent → Function:*

$$m_{i \to g}(a_i) = \sum_{h \in \mathcal{F}_i \setminus \{g\}} m_{h \to i}(a_i) + c_{ig}.$$

(2)

*Function → Agent:*

$$m_{g \to i}(a_i) = \max_{\boldsymbol{a}_g \setminus a_i} \left[ q(\boldsymbol{a}_g; z_t) + \sum_{h \in \mathcal{V}_g \setminus \{i\}} m_{h \to g}(a_h) \right].$$

(3)

This embedding-aware mechanism adapts edge evaluation and sparsification to evolving task demands. Further details on the message definitions and normalization are provided in Appendix B.2.

**Coordination-Aware Signal Extraction.** After $T$ rounds of message passing, we compute a coordination-aware action score for each agent:

$$a_i^* = \arg\max_{a_i} \sum_{g \in \mathcal{F}_i} m_{g \to i}(a_i). \tag{4}$$

The resulting Max-Sum messages are used only as coordination-aware signals to augment local observations, rather than to select executed actions. In practice, $T$ is treated as a hyperparameter. Unless otherwise stated, we use $T = 5$ rounds of Max-Sum message passing in all experiments, following prior sparse coordination-graph implementations. If $\mathcal{G}$ is large or changes dynamically, the same message-passing procedure can be rerun after the graph is updated, initializing messages with previous iterations to speed up convergence.

## 2.2. Multi-Task Dec-POMDP Extension

We extend the standard Dec-POMDP formulation to a multi-task regime, denoted as a multi-task Dec-POMDP:

$$\mathcal{T} = \langle I, S, A, P, R, \Omega, O, n, \gamma, \mathcal{K} \rangle, \tag{5}$$

where $\mathcal{K}$ is a finite set of tasks and each task $k \in \mathcal{K}$ defines a sub-environment $\mathcal{T}_k$. The global state and observation spaces are unions over task-specific domains, i.e., $S = \bigcup_k S_k$ and $\Omega = \bigcup_k \Omega_k$.

Unlike conventional multi-task RL, where the task identity $k$ is given, we adopt an *unsupervised multi-task setting* in which $k$ remains hidden, and agents must infer latent task types from their trajectory histories. This reflects realistic MARL domains (e.g., competitive games or decentralized control) where agents must adapt to diverse tasks without explicit task labels. Full details of support differences are provided in Appendix B.3 and B.4.

## 2.3. VQ-VAE for Quantized Latent Space Generation

To obtain compact discrete representations of trajectory segments in multi-task MARL, we adopt a VQ-VAE (Li & Yuan, 2021) as extended in LAGMA (Na & Moon, 2024). Given a segment $s$, the encoder maps it to a continuous vector and quantizes it via the nearest codebook entry:

$$x = f_\phi^e(s), \; x_q = [x]_q = e_z, \; z = \arg\min_j \|x - e_j\|_2. \tag{6}$$

The quantized embeddings are trained with a VQ-VAE loss augmented by a coverage regularization term to encourage broader codebook usage across tasks:

$$\mathcal{L}_{\text{VQ}}^{\text{tot}} = \mathcal{L}_{\text{VQ}} + \lambda_{\text{cvr}} \cdot \mathcal{L}_{\text{cvr}}. \tag{7}$$

This promotes active utilization of the full codebook, ensuring the latent space retains sufficient capacity for clustering and generalization (see Appendix B.5 for full derivations). For a detailed discussion of related work, see Appendix A.

## 3. Proposed Scheme

By bridging trajectory-level abstraction with interaction-level adaptation, our method explicitly couples latent task representations with dynamic agent relations in a unified CTDE MARL framework. (see Figure 1).

### 3.1. Dynamic Sparse Coordination Graph Construction

To improve coordination efficiency and scalability, we construct dynamic coordination graphs that adapt to evolving task contexts and agent interactions. At graph-update steps, inspired by recent sparse and context-aware graph methods, we predict the latent task cluster $k \in \{1, \ldots, K\}$ using VQ-VAE-based representations of agents' trajectories $\tau_i$. For each cluster $k$, the edge importance between agents $i$ and $j$ is defined as:

$$\zeta_{ij}^{(k)} = \text{Var}_{a_j \sim \pi_j} \left( q_{ij}(\tau_{ij}, a_i, a_j) \mid k \right), \tag{8}$$

where $\zeta_{ij}^{(k)}$ is an action-marginal sensitivity proxy estimated from rollout mini-batches. Following sparse coordination-graph theory, lower payoff variance means an edge is less likely to affect greedy action selection, so variance is used as a practical relevance proxy. In TACTIC, $\zeta_{ij}^{(k)}$ is computed within latent task cluster $k$, and the top-$r$ edges are retained to form a sparse task-specific coordination graph (Appendix D.7). Here, $k$ denotes the trajectory class inferred from the VQ-VAE embedding and classifier.

#### 3.1.1. TASK-CONDITIONED LOW-RANK PAYOFF FACTORIZATION

To enhance parameter efficiency and task-aware generalization, each pairwise payoff function $q_{ij}$ is decomposed via a rank-$R$ factorization conditioned on the task embedding $e_k \in \mathbb{R}^{d_e}$ (derived from the latent $z_t$):

$$q_{ij}(a_i, a_j, k) = \sum_{r=1}^{R} L_r(a_i, e_k) \cdot R_r(a_j, e_k), \tag{9}$$

where $L_r$ and $R_r$ are learnable task-conditioned basis functions. This provides a parameter-efficient representation of pairwise payoffs, reducing the payoff parameterization from $O(|\mathcal{A}|^2)$ to $O(R|\mathcal{A}|)$, while retaining compatibility with Max-Sum-style coordination updates and enabling flexible payoff adaptation across tasks.

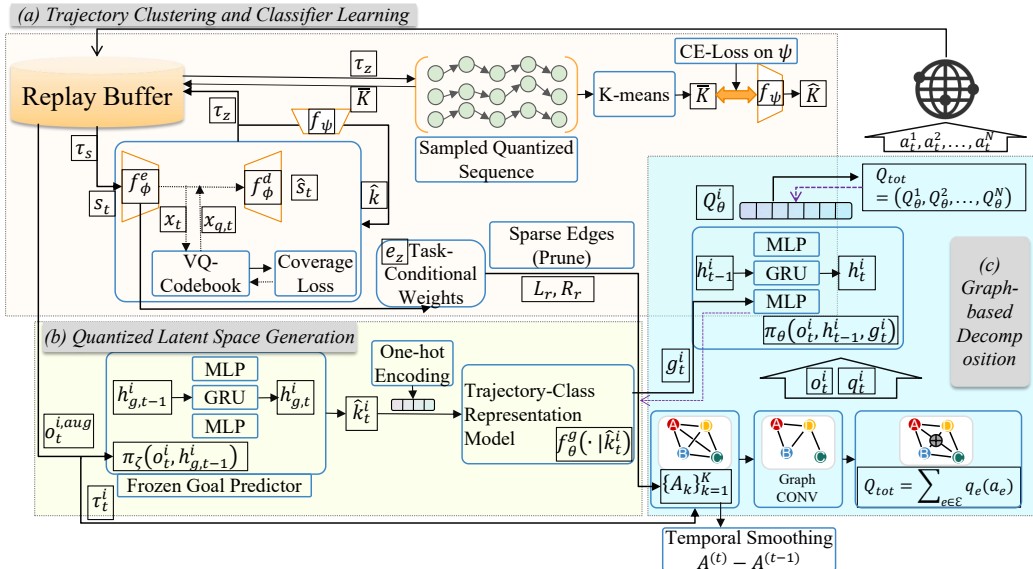

*Figure 1.* (a) Trajectory Clustering and Classification: Local trajectories are encoded and quantized via VQ-VAE, then clustered into discrete trajectory classes using K-means. An auxiliary classifier provides online class prediction. (b) Quantized Latent Space: Coverage loss enforces even utilization of codebook vectors across tasks, preventing codebook collapse and improving cluster separability. (c) Graph-based Decomposition: Trajectory-class embeddings condition the construction of dynamic coordination graphs with sparse, task-weighted edges. Temporal smoothing regularizes adjacency changes, and sparse pruning improves scalability. Frozen Goal Predictor: The goal predictor is frozen during training to prevent unstable high-level signals from disrupting policy learning, enhancing stability and generalization.

### 3.1.2. SYMMETRIC PAYOFF AND ACTION EMBEDDING

To support reciprocal coordination (e.g., attacker–defender switches), we enforce symmetric payoff evaluation:

$$q_{ij}^{\text{sym}}(a_i, a_j) = \tfrac{1}{2}\big(q_{ij}(a_i, a_j) + q_{ji}(a_j, a_i)\big), \qquad (10)$$

which prevents directional bias and ensures consistent coordination under role exchanges. This symmetricization assumes compatible action spaces for $i$ and $j$; when heterogeneous, we apply it over a shared embedding.

### 3.1.3. TASK-AWARE QUANTIZATION AND GRAPH GENERATION

We introduce a VQ-VAE quantization layer that maps history embeddings $h_i^t$ into a discrete task codebook $\{e_k\}_{k=1}^K$:

$$z_i^t = \text{Quantize}\big(h_i^t, \{e_k\}\big), \qquad (11)$$

where $z_i^t$ denotes agent $i$'s quantized trajectory code. The codebook is updated by the VQ-VAE objective, while an additional entropy regularizer encourages diverse class usage:

$$\mathcal{L}_{\text{ent}} = -\mathcal{H}\big(p(k|\tau_i)\big) = \sum_k p(k|\tau_i) \log p(k|\tau_i). \qquad (12)$$

Task-conditioned embeddings $z_i^t$ are then used to parameterize edge scores and generate a time-varying coordination graph:

$$\boldsymbol{A}_{ij}^{(t)} = \text{Softmax}\big(W_\phi[z_i^t; z_j^t]\big). \qquad (13)$$

To encourage temporal stability, we apply

$$\mathcal{L}_{\text{smooth}} = \sum_{t=2}^{T} \|\boldsymbol{A}^{(t)} - \boldsymbol{A}^{(t-1)}\|_F^2. \qquad (14)$$

The semantic adjacency scores provide task-conditioned edge weights, while the variance-based top-$r$ pruning described above determines the retained sparse topology.

### 3.2. Trajectory-Class-Aware Policy

To enable task generalization, we discretize history sequences into quantized latent sequences using a modified VQ-VAE, following TRAMA (Na et al., 2025) and LAGMA (Na & Moon, 2024). K-means is used to obtain trajectory-class centroids, which are stabilized by reusing previous centroids across clustering updates; a trajectory-class predictor then provides class-conditioned inputs to agents. The goal predictor is frozen during training to prevent gradient interference and maintain stability. Both encoder and decoder are two-layer MLPs with ReLU activations, and we maintain both a hard code and a soft posterior $p(k \mid \tau)$ for class-conditioning.

### 3.2.1. CLASS-CONDITIONAL VQ-VAE AND COVERAGE LOSS

To prevent codebook collision in multi-task settings, we adopt a class-conditional VQ-VAE that jointly optimizes

task-specific code vectors and coverage consistency. The coverage regularization is defined as:

$$\mathcal{L}_{\mathrm{cvr}}(e, k) = \frac{1}{|\mathcal{J}(t, k)|} \sum_{j \in \mathcal{J}(t, k)} \left\| \mathrm{sg}\left[ f_\phi^e \left( s_t^k \right) \right] - e_j \right\|_2^2, \tag{15}$$

where $\mathcal{J}(t, k)$ indexes the subset of codebook entries assigned to timestep $t$ and trajectory class $k$, and $s_t^k$ denotes a state sampled from trajectories assigned to class $k$. The final training loss is:

$$\mathcal{L}_{\mathrm{VQ}}^{\mathrm{tot}}(\phi, \boldsymbol{e}) = \mathcal{L}_{\mathrm{VQ}}(\phi, \boldsymbol{e}) + \lambda_{\mathrm{cvr}} \mathcal{L}_{\mathrm{cvr}} + \lambda_{\mathrm{ent}} \mathcal{L}_{\mathrm{ent}}. \tag{16}$$

Here, $\mathcal{L}_{\mathrm{VQ}}$ is the standard VQ-VAE objective, which already includes the reconstruction, vector-quantization, and commitment terms. The entropy term $\mathcal{L}_{\mathrm{ent}} = -\mathcal{H}(p(k|\tau))$ encourages diverse class usage, while $\mathcal{L}_{\mathrm{cvr}}$ promotes balanced codebook utilization across trajectory classes.

### 3.2.2. TRAJECTORY CLUSTERING AND CLASSIFIER LEARNING

To distinguish trajectory patterns across tasks, we periodically cluster latent trajectory embeddings to identify task-consistent classes. Given a trajectory sequence $\tau_{s_{t=0}}$, we encode it using the trained VQ-VAE encoder into quantized codes $x_{q,t} \in \mathbb{R}^{d_e}$ with indices $z_t \in \{1, \ldots, K\}$ from the shared codebook $\{e_j\}_{j=1}^K$. Each trajectory embedding is represented as $\tau_{\mathcal{Z}_{t=0}} = \{z_0, z_1, \ldots, z_T\}$, and its compact representation is obtained by summing codebook vectors:

$$\bar{e}^m = \sum_{t=0}^{T} e_{j=z_t}^m. \tag{17}$$

K-means clustering is performed on $\{\bar{e}^m\}_{m=1}^M$. In the main experiments, $n_{\mathrm{cl}}$ is selected from a small candidate set using silhouette scores and then kept fixed across tasks; we further report sensitivity to $n_{\mathrm{cl}}$ in the appendix. To maintain label stability across iterations, previous centroids are reused for initialization, reducing permutation and class switching.

Between clustering updates, an auxiliary classifier $f_\psi(\cdot | \bar{e}^m)$ predicts trajectory classes online using cross-entropy loss:

$$\mathcal{L}_{\mathrm{cls}} = -\frac{1}{M} \sum_{m=1}^{M} \log f_\psi\left( \bar{k}_m \mid \bar{e}_m \right), \tag{18}$$

where $\bar{k}_m$ is the cluster label. Predicted trajectory classes $\hat{k}_m$ provide task-semantic inputs for graph modulation. The classifier is updated only in the representation stage, while its policy-side copy is frozen as the goal predictor during RL control.

### 3.2.3. POLICY CONDITIONING ON TRAJECTORY CLASS

Inspired by TRAMA (Na et al., 2025), we adopt a trajectory-class-aware policy that conditions agent decisions on local observations and predicted trajectory goals. Unlike TRAMA, we freeze the goal predictor and remove its training loss to avoid unstable task-prediction gradients.

At each timestep $t$, the policy of agent $i$ is defined as

$$\pi_i(a_t^i \mid o_t^{i,\mathrm{aug}}, g_t^i), \tag{19}$$

where $o_t^{i,\mathrm{aug}} = [o_t^i; q_t^i]$ combines the local input with coordination-aware signals from the sparse graph module, and $g_t^i$ is a discrete trajectory-class label.

A lightweight goal predictor estimates trajectory goals from unaugmented observations:

$$p(g_t^i \mid o_t^i) = \mathrm{GoalPredictor}(o_t^i), \quad g_t^i = \arg\max_k p(k \mid o_t^i), \tag{20}$$

implemented as a two-layer MLP with softmax output over $n_{cl}$ classes. All goal predictor parameters $\theta_g$ are frozen during training.

See Appendix D.3 for further ablation and visualization of label consistency and training stability.

### 3.3. Overall Learning Objective

We adopt the QMIX (Rashid et al., 2018) value factorization framework to train individual agent value functions $Q_\theta^i$ via a mixing network $Q_\theta^{\mathrm{tot}}$, which aggregates all $Q_i$ into a monotonic global value to ensure decentralized execution. Each agent's behavior is conditioned on its coordination-aware observation and the predicted trajectory-class goal $\boldsymbol{g}$, generated by a pretrained and frozen goal predictor $f_{\theta_g}^g$. Executed actions are produced by the local policy

$$\pi_i(a_t^i \mid o_t^{i,\mathrm{aug}}, g_t^i), \tag{21}$$

where the sparse coordination-graph / Max-Sum module provides coordination-aware signals used in $o_t^{i,\mathrm{aug}}$, while QMIX is used only for centralized TD training under CTDE. In TACTIC, we exclude $\theta_g$ from RL optimization to prevent catastrophic forgetting due to unstable task predictions. A component-wise comparison with TRAMA is provided in Appendix D.8. The RL objective minimizes the temporal-difference (TD) error between current and target estimates:

$$\mathcal{L}_{\mathrm{TD}}(\theta) = \mathbb{E}_{(\boldsymbol{o}, \boldsymbol{a}, r, \boldsymbol{o}') \sim \mathcal{D}} \Big[ \big( r + \gamma \max_{\boldsymbol{a}'} Q_{\theta^-}^{\mathrm{tot}}(\boldsymbol{o}', \boldsymbol{g}', \boldsymbol{a}') \\ - Q_\theta^{\mathrm{tot}}(\boldsymbol{o}, \boldsymbol{g}, \boldsymbol{a}) \big)^2 \Big], \tag{22}$$

where $\boldsymbol{g} = \mathrm{sg}(f_{\theta_g}^g(\boldsymbol{o}))$ and $\boldsymbol{g}' = \mathrm{sg}(f_{\theta_g}^g(\boldsymbol{o}'))$. In the representation-update stage, we optimize the VQ-VAE quantization loss $\mathcal{L}_{\mathrm{VQ}}^{\mathrm{tot}}(\phi, \boldsymbol{e})$ and the trajectory classification loss

$\mathcal{L}_{\text{cls}}$ defined in Eq. (18). During RL control training, the policy-side goal predictor is frozen. We therefore separate representation learning from RL control:

$$\mathcal{L}_{\text{rep}} = \lambda_{\text{VQ}}\mathcal{L}_{\text{VQ}}^{\text{tot}} + \lambda_{\text{cls}}\mathcal{L}_{\text{cls}}, \tag{23}$$

$$\mathcal{L}_{\text{RL}} = \mathcal{L}_{\text{TD}}(\theta) + \lambda_{\text{smooth}}\mathcal{L}_{\text{smooth}}. \tag{24}$$

Here, $\mathcal{L}_{\text{rep}}$ updates the VQ-VAE and classifier, whereas $\mathcal{L}_{\text{RL}}$ updates the policy and graph module with the goal predictor frozen. The pseudocode is shown in the Appendix C.1. See Appendices D.4 and D.9 for engineering analyses of runtime efficiency, communication cost, and graph-update policies.

# 4. Experiments

## 4.1. Experimental Setups

We evaluate TACTIC on the StarCraft II multi-agent benchmark (2–27 agents, 5 seeds) and on SUMO traffic control tasks (8 agents, long-duration). For fairness, we closely follow TRAMA's setup, with minor adjustments for stability. Details of hyperparameter modifications and their impact are provided in Appendix D.1.

## 4.2. StarCraft II and Test Performance

### 4.2.1. ENVIRONMENTS AND TASKS

We evaluate TACTIC on SMACv1/v2 and a multi-task SMAC suite with diverse unit compositions (Ellis et al., 2023)(Na & Moon, 2024), map layouts, and dynamic enemy behaviors. Task identities are hidden during training and evaluation, enabling a rigorous assessment of trajectory-aware adaptation and structure-sensitive coordination under unsupervised multi-task settings.

### 4.2.2. RESULTS ON SMACv1

As shown in Table 1, TACTIC achieves strong overall performance across the evaluated SMACv1 maps. Compared with subtask-based methods such as RODE and TRAMA, TACTIC obtains competitive or higher returns and win rates. Compared with value-decomposition and memory-based baselines, TACTIC shows more favorable performance in these sparse-reward coordination tasks. Compared with graph-based methods such as DMCG and LTSCG, TACTIC remains competitive while introducing task-conditioned structural adaptation.

### 4.2.3. RESULTS ON SMACv2

**Baseline comparisons.** As shown in Figure 2, TACTIC achieves the highest win rate and the most stable learning curve among all baselines. Compared to TRAMA and DMCG, which show moderate results, and memory-

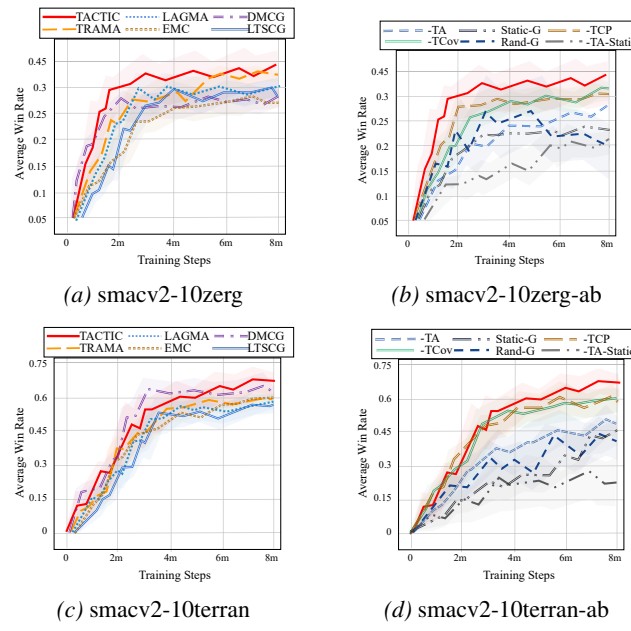

*(a)* smacv2-10zerg      *(b)* smacv2-10zerg-ab

*(c)* smacv2-10terran      *(d)* smacv2-10terran-ab

*Figure 2.* StarCraft II average win-rate curves of baselines and ablations (SMACv2).

based methods (LAGMA, EMC) that perform poorly, TACTIC consistently generalizes better under challenging high-coordination settings.

**Ablation studies.** **-TA**: Removing trajectory abstraction slows convergence and reduces final performance. **-TCov**: Removing coverage loss decreases stability, showing its role in exploration. **Static-G vs Rand-G**: Static-G performs reasonably well in simple dynamics, but Rand-G introduces erratic training due to random connectivity. **-TCP and -TA-Static**: Both yield the worst results, confirming that trajectory classification and graph modulation must act synergistically to handle multi-task complexity.

### 4.2.4. RESULTS ON SURCOMB BENCHMARKS

We evaluate TACTIC and its ablations on SurComb3/4 and their mirrored variants (reSurComb3/4) (Na et al., 2025) as shown in Figure 3, multi-task benchmarks derived from SMACv2 with diverse unit compositions, stochastic positions, and sparse rewards. No explicit task ID is given during training or evaluation, enforcing unsupervised multi-task generalization. TRAMA is used as the sole baseline, as it previously achieved state-of-the-art performance in these settings.

Results show that TACTIC performs strongly in standard SurComb3/4 tasks, while the mirrored reSurComb variants reveal an important semantic-alignment boundary case. In these variants, Static-G can surpass full TACTIC in final returns, and Rand-G shows unstable but occasionally near-optimal performance, suggesting that dynamic graph adap-

*Table 1.* Performance comparison on SMACv1 5m_vs_6m, 8m_vs_9m, MMM2, and 27m_vs_30m, grouped by algorithm class. Mean episode return and win rate.

| Algorithm | 5m_vs_6m | | 8m_vs_9m | | MMM2 | | 27m_vs_30m | |
|---|---|---|---|---|---|---|---|---|
| | Reward | Win Rate | Reward | Win Rate | Reward | Win Rate | Reward | Win Rate |
| TACTIC | **14.50 ± 0.75** | **0.46 ± 0.16** | **18.16 ± 0.10** | **0.78 ± 0.19** | **19.22 ± 2.37** | **0.89 ± 0.25** | **14.67 ± 1.21** | **0.46 ± 0.13** |
| **Subtask-based** | | | | | | | | |
| RODE | 12.10 ± 1.05 | 0.26 ± 0.01 | 15.30 ± 1.00 | 0.44 ± 0.08 | 16.80 ± 2.50 | 0.78 ± 0.15 | 12.90 ± 1.50 | 0.36 ± 0.10 |
| LIIR | 7.50 ± 1.30 | 0.13 ± 0.04 | 8.20 ± 1.25 | 0.10 ± 0.02 | 9.00 ± 2.10 | 0.13 ± 0.02 | 6.00 ± 1.80 | 0.10 ± 0.05 |
| TRAMA | 13.70 ± 0.80 | 0.37 ± 0.17 | 18.09 ± 0.50 | 0.76 ± 0.13 | 19.16 ± 2.53 | 0.85 ± 0.18 | 14.15 ± 1.67 | 0.45 ± 0.11 |
| **Value-Decomposition** | | | | | | | | |
| QMIX | 11.20 ± 1.10 | 0.31 ± 0.12 | 7.50 ± 1.20 | 0.08 ± 0.01 | 8.30 ± 1.80 | 0.12 ± 0.01 | 5.20 ± 1.50 | 0.08 ± 0.02 |
| QTRAN | 11.80 ± 1.00 | 0.33 ± 0.29 | 10.20 ± 1.00 | 0.26 ± 0.02 | 7.00 ± 2.10 | 0.03 ± 0.01 | 4.50 ± 1.30 | 0.03 ± 0.02 |
| QPLEX | 11.90 ± 0.95 | 0.34 ± 0.21 | 16.50 ± 0.70 | 0.65 ± 0.21 | 10.50 ± 1.90 | 0.25 ± 0.05 | 8.20 ± 1.50 | 0.22 ± 0.06 |
| GraphMix | 10.30 ± 1.20 | 0.31 ± 0.18 | 13.00 ± 1.10 | 0.28 ± 0.16 | 5.10 ± 1.50 | 0.01 ± 0.01 | 3.50 ± 1.20 | 0.01 ± 0.01 |
| QATTEN | 11.50 ± 1.10 | 0.35 ± 0.16 | 12.50 ± 1.00 | 0.27 ± 0.13 | 10.20 ± 1.40 | 0.21 ± 0.02 | 7.80 ± 1.50 | 0.21 ± 0.03 |
| **Memory-based** | | | | | | | | |
| EMC | 9.50 ± 1.30 | 0.24 ± 0.14 | 10.20 ± 1.10 | 0.23 ± 0.15 | 7.90 ± 1.70 | 0.15 ± 0.04 | 5.00 ± 1.40 | 0.12 ± 0.04 |
| LAGMA | 12.30 ± 1.00 | 0.32 ± 0.23 | 17.00 ± 0.90 | 0.67 ± 0.26 | 16.10 ± 2.10 | 0.68 ± 0.20 | 12.00 ± 1.60 | 0.32 ± 0.10 |
| **Coordinate Relationships** | | | | | | | | |
| G2ANet | 12.00 ± 0.90 | 0.34 ± 0.14 | 16.20 ± 0.80 | 0.61 ± 0.17 | 15.20 ± 2.00 | 0.60 ± 0.18 | 12.20 ± 1.50 | 0.36 ± 0.11 |
| LTSCG | 12.20 ± 0.85 | 0.35 ± 0.25 | 17.50 ± 0.70 | 0.72 ± 0.03 | 16.00 ± 1.90 | 0.70 ± 0.15 | 13.00 ± 1.40 | 0.40 ± 0.12 |
| DICG | 13.00 ± 0.75 | 0.42 ± 0.38 | 16.00 ± 0.80 | 0.63 ± 0.24 | 16.20 ± 2.00 | 0.65 ± 0.20 | 13.40 ± 1.30 | 0.41 ± 0.13 |
| DMCG | 13.40 ± 0.65 | 0.45 ± 0.18 | 17.80 ± 0.60 | 0.75 ± 0.13 | 17.30 ± 2.20 | 0.78 ± 0.16 | 14.20 ± 1.20 | 0.44 ± 0.12 |

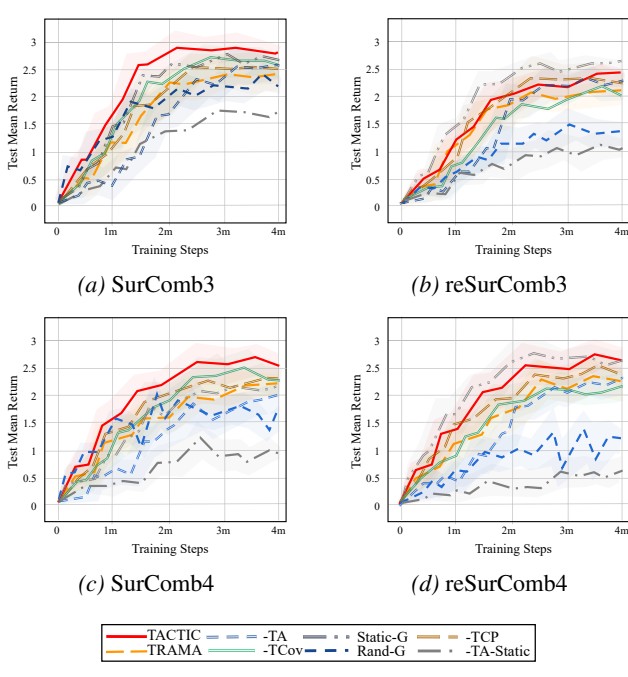

*(a)* SurComb3  *(b)* reSurComb3

*(c)* SurComb4  *(d)* reSurComb4

| | | |
|---|---|---|
| TACTIC | -TA | Static-G | -TCP |
| TRAMA | -TCov | Rand-G | -TA-Static |

*(e)* SurComb task illustration

*Figure 3.* Test mean return curves of baselines and ablations on SurComb and reSurComb benchmarks.

tation is most reliable when latent trajectory semantics remain aligned with coordination structure. These outcomes highlight that trajectory-conditioned graphs are generally effective, but in semantically altered tasks, a simpler static graph can sometimes be a more reliable fallback when the latent semantic signal is less aligned with the coordination structure. For detailed ablation analyses on SMAC and hyperparameter robustness, please refer to Appendix D.2.

### 4.3. SUMO Environment and Evaluation

#### 4.3.1. SCENARIO CONFIGURATION

We evaluate TACTIC on the SUMO (Lopez et al., 2018) traffic benchmark, following the cooperative adaptive vehicle (CAV) setup of Guo et al. (Guo et al., 2024). The environment is a four-way intersection with mixed autonomy traffic (300 vehicles/hour/lane) including eight controlled CAVs. Each CAV aims to cross the intersection safely while maximizing traffic efficiency. Task variability arises from randomized initial positions, routes, and communication conditions.

To test robustness, we introduce a 600 ms vehicle-to-vehicle (V2V) delay during testing for zero-shot transfer evaluation. All model, observation, and reward specifications follow (Guo et al., 2024) and are detailed in Appendix D.6.

#### 4.3.2. PERFORMANCE METRICS AND RESULTS

Figure 4 reports quantitative performance on the non-signalized intersection task. Four metrics are used for performance evaluation: **(1)** *Mean Return*: Cumulative reward over episode, capturing overall policy quality; **(2)** *Completion Rate*: Fraction of episodes where all CAVs safely

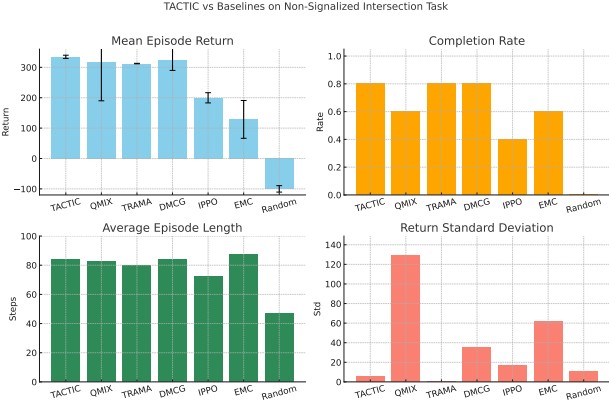

*Figure 4.* Performance comparison across TACTIC and baseline methods on the non-signalized intersection task. From left to right: (a) average episode return with standard deviation, (b) task completion rate, (c) average episode length, and (d) return variance. TACTIC achieves the best performance across all metrics, with high return, stable training, and efficient coordination.

cross the intersection; **(3)** *Episode Length*: Number of steps required to complete or terminate an episode, reflecting traffic efficiency together with completion rate; **(4)** *Return Std*: Measures policy stability—lower values indicate more consistent decision-making. The comprehensive evaluation in Table 2 and Figure 4 reveals distinct performance patterns across different MARL paradigms when applied to the non-signalized intersection task.

**Results on SUMO.** Based on Figure 4, TACTIC obtains the highest average return, the highest completion rate, and the lowest return variance among the compared methods. These results suggest favorable coordination efficiency and decision stability in the non-signalized intersection task. TRAMA and DMCG also perform competitively, with slightly lower returns and higher variance. QMIX achieves moderate return and completion rate, while EMC and IPPO show weaker performance under sparse rewards and mixed-autonomy coupling. The Random baseline fails to complete the task, indicating the need for learned coordination policies in this setting.

*Table 2.* Deployment-oriented environmental comparison between TACTIC and the Random baseline.

| Metric | TACTIC | Random |
|--------|--------|--------|
| Arrival Time (s) | **15.90** | 18.60 |
| Time Loss (s) | **1.48** | 1.59 |
| CO (mg) | **786.92** | 799.56 |
| $CO_2$ (mg) | **59596.46** | 60983.86 |
| HC (mg) | **4.77** | 4.98 |
| PMx (mg) | **0.89** | 1.21 |
| NOx (mg) | **22.94** | 25.63 |
| Fuel (mg) | **19008.64** | 21813.26 |

Based on Table 2, TACTIC yields lower fuel consumption

and pollutant emissions than the Random baseline, suggesting that improved coordination can translate into favorable downstream environmental indicators. Our results demonstrate that while TACTIC introduces moderate overhead during training, its inference-time efficiency is comparable to lighter baselines. For complete wall-clock time, memory, and CPU/GPU usage statistics across all methods, see Appendix D.4.

## 5. Conclusion

In this work, we proposed TACTIC, a unified CTDE framework for adaptive coordination in cooperative multi-agent reinforcement learning (MARL). TACTIC addresses multi-task MARL by using trajectory-conditioned task semantics to construct dynamic sparse coordination graphs, enabling agents to adapt their interaction structure across changing task contexts under sparse rewards. TACTIC uses a VQ-VAE-based trajectory encoder to infer discrete latent classes that serve as implicit task semantics. These latent representations condition graph construction and payoff modulation, allowing the coordination topology to adapt to evolving inter-agent dependencies. A temporal smoothing loss further stabilizes graph adaptation over sequential decision steps. TACTIC works best when latent trajectory semantics remain aligned with coordination structure, which defines its intended scope (Appendix E). Experiments on SMACv1/v2 and SUMO show that TACTIC achieves strong overall performance in both single-task and generalized multi-task settings. Compared with competitive baselines such as TRAMA, DMCG, and LAGMA, TACTIC shows strong overall competitiveness in episodic return, convergence, and variance; the SUMO deployment-oriented analysis further suggests favorable graph sparsity, runtime efficiency, and downstream environmental indicators relative to the uncoordinated baseline.

## Acknowledgements

This work was supported in part by the National Natural Science Foundation of China under Grant Nos. 62372243, 624B2101, and 62422605, and in part by the China Postdoctoral Science Foundation under Grant No. 2025M781476.

## Impact Statement

This paper presents work whose goal is to advance the field of Machine Learning. There are many potential societal consequences of our work, none which we feel must be specifically highlighted here.

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

## A. Related Work

**Multi-Agent Modeling and Prediction:** Beyond single-agent abstraction, many multi-agent systems (MAS) require modeling teammates or opponents (Sun et al., 2025). Early studies estimated opponents' values or policies under explicit state assumptions (He et al., 2016), while later works inferred hidden states to predict actions (Raileanu et al., 2018) and introduced VAE-based trajectory inference under limited communication (Papoudakis & Albrecht, 2020; Papoudakis et al., 2021). Recent advances in MARL move toward group-level roles and multi-task coordination: RODE learns role-based decompositions for scalability (Wang et al., 2021b), LIIR optimizes individual intrinsic rewards to separate agent contributions (Du et al., 2019), and TRAMA leverages trajectory-class awareness to infer task types and adapt policies (Na et al., 2025). Unlike methods assuming fixed coordination or focusing only on action prediction, our approach infers task context and adapts coordination dynamically over time without explicit task labels or execution-time communication. The resulting latent-conditioned, time-varying graph explicitly captures structural adaptation across tasks.

**Abstraction of States and Trajectories for Sparse-Reward MARL:** State and trajectory abstraction has long been used to scale RL to large domains, from early discretization and model-based grouping by dynamics or rewards (Grześ & Kudenko, 2008; Dasgupta, 2000; Zhu et al., 2021) to model-free variants that remove irrelevant features or discretize large spaces (Tang et al., 2017). NECSA extends this idea with grid-based state abstraction for episodic control but remains inefficient in continuous or high-dimensional domains (Li et al., 2023). More recent work learns semantic embeddings: EMC introduces curiosity-driven episodic memory for coordinated exploration (Zheng et al., 2021), while LAGMA and TRAMA use VQ-VAE to discretize trajectories for latent goal discovery and cross-task transfer (Na & Moon, 2024; Li & Yuan, 2021). Building on this line, our framework employs VQ-VAE embeddings to capture latent task semantics and dynamically reshape coordination among agents. By clustering structurally related trajectories, it enhances reward propagation under sparse feedback and enables fast adaptation to new tasks while preserving prior knowledge.

**Coordination-Graph and Value-Factorized MARL: Factorized Value-Based MARL.** Early MARL advances decomposed the joint $Q$ into per-agent components (Rashid et al., 2018; Foerster et al., 2018; Son et al., 2019; Wang et al., 2021a), mitigating non-stationarity and working well on simpler tasks. However, under partial observability or stochasticity, purely local utilities often miscoordinate because they cannot separate cooperative from conflicting behaviors (Sun et al., 2023), leading to *relative overgeneralization* (Panait et al., 2006). Moreover, context-agnostic factorization cannot adapt inter-agent dependencies as task structure changes, a growing limitation in multi-task or dynamic settings.

**Coordination Graph Extensions and Higher-Order Dependencies.** Coordination graphs (CGs) capture dependencies without full joint enumeration by modeling agents as nodes with payoff edges (Kok & Vlassis, 2006; Peng et al., 2024); deep extensions alleviate overgeneralization in large/continuous domains (Boehmer et al., 2020). Subsequent work learns sparse or adaptive graph structures using payoff variance or implicit attention mechanisms (Wang et al., 2022; Li et al., 2021), and explores non-linear utilities (Kang et al., 2022) or hyper-edges for multi-agent outcomes (Castellini et al., 2019). Real settings also involve indirect or multi-step influences: DMCG models such indirect dependencies via emergent paths (Gupta et al., 2026), but does not explicitly couple graph adaptation with trajectory-level task semantics; LTSCG builds sparse graphs from history with "predict-future / infer-present" mechanisms (Duan et al., 2024), yet it does not explicitly inject task semantics into graph construction, constraining generalization in multi-task or semantically diverse environments.

Building on these insights, we propose a *task-conditioned dynamic coordination graph* that integrates VQ-VAE latent codes with edge gating, enabling structure-aware generalization across dynamic multi-agent tasks.

## B. Additional Preliminaries and Detailed Formulations

### B.1. Detailed Coordination Graph Formulation

A coordination graph $\mathcal{G} = \langle \mathcal{V}, \mathcal{E} \rangle$ represents agents as vertices $v_i \in \mathcal{V}$, where each edge $\{i,j\} \in \mathcal{E}$ captures crucial coordination dependencies. In the pairwise case, the global $Q$-function can be decomposed as:

$$Q_{\text{tot}}(\boldsymbol{\tau}, \boldsymbol{a}) = \frac{1}{|\mathcal{V}|} \sum_i q_i (\tau_i, a_i) + \frac{1}{|\mathcal{E}|} \sum_{\{i,j\} \in \mathcal{E}} q_{ij} (\tau_i, \tau_j, a_i, a_j). \tag{25}$$

Here: $q_i(\tau_i, a_i)$ denotes the per-agent utility, $q_{ij}(\tau_i, \tau_j, a_i, a_j)$ denotes the payoff for agent pair $(i, j)$, $\tau_i$ is agent $i$'s local history, and $\tau_{ij} = \langle \tau_i, \tau_j \rangle$.

The division by $|\mathcal{V}|$ or $|\mathcal{E}|$ is illustrative; in practice, different terms may be weighted unequally. Traditional approaches typically assume $\mathcal{G}$ is static, but this limits adaptability when task semantics or coordination structures change.

## B.2. Detailed Max-Sum Message Passing Formulation

In our task-adaptive coordination graph $\mathcal{G}_t = \langle \mathcal{V}_a, \mathcal{V}_q, \mathcal{E}_t \rangle$, $\mathcal{V}_a$ denotes agent (variable) nodes for each action $a_i$, $\mathcal{V}_q$ denotes function nodes for local utilities $q_i$ or payoffs $q_{ij}$, and $\mathcal{E}_t$ is the dynamic edge set. At each timestep, $\mathcal{E}_t$ is constructed from VQ-VAE-based trajectory embeddings $z_t$.

For an agent node $i$, let $\mathcal{F}_i$ be the set of adjacent function nodes. The message to a function node $g \in \mathcal{F}_i$ is:

$$m_{i \to g}(a_i) = \sum_{h \in \mathcal{F}_i \setminus \{g\}} m_{h \to i}(a_i) + c_{ig}, \tag{26}$$

where $c_{ig}$ optionally normalizes or re-centers messages.

For a function node $g$, let $\mathcal{V}_g$ denote its adjacent agent nodes, and $\mathbf{a}_g$ their joint action. The message to agent $i \in \mathcal{V}_g$ is:

$$m_{g \to i}(a_i) = \max_{\mathbf{a}_g \setminus a_i} \Big[ q(\mathbf{a}_g; z_t) + \sum_{h \in \mathcal{V}_g \setminus \{i\}} m_{h \to g}(a_h) \Big]. \tag{27}$$

Here $q(\mathbf{a}_g; z_t)$ is a context-aware payoff function whose parameters are conditioned on latent embedding $z_t$. This enables both the parameterization of $q_{ij}$ and the dynamic structure of $\mathcal{G}_t$ to reflect latent task semantics.

## B.3. Detailed Multi-Task Dec-POMDP Formulation

A multi-task Dec-POMDP is defined as $\mathcal{T} = \langle I, S, A, P, R, \Omega, O, n, \gamma, \mathcal{K} \rangle$, where $\mathcal{K}$ is a finite set of tasks. Each task $k \in \mathcal{K}$ corresponds to a sub-environment $\mathcal{T}_k = \langle I, S_k, A, P, R, \Omega_k, O, n, \gamma \rangle$. The global state and observation spaces are unions of task-specific domains: $S = \bigcup_k S_k$, $\Omega = \bigcup_k \Omega_k$, where $\Omega_k$ is the task-specific observation space.

## B.4. Support Differences in Multi-Task Dec-POMDPs

While tasks share a unified API for the transition, reward, and observation functions, we model task-indexed kernels $(P, R, O)$ whose supports can be disjoint across tasks:

$$\begin{aligned}
\mathrm{dom}(P_{k_1}) \setminus \mathrm{dom}(P_{k_2}) &\neq \emptyset, \\
\mathrm{dom}(R_{k_1}) \setminus \mathrm{dom}(R_{k_2}) &\neq \emptyset, \\
\mathrm{dom}(O_{k_1}) \setminus \mathrm{dom}(O_{k_2}) &\neq \emptyset.
\end{aligned} \tag{28}$$

Here, $\mathrm{dom}(P)$ denotes the domain of the transition function, i.e., the set of all valid state-action pairs. As an example, two SMACv2 scenarios with different unit compositions span distinct subspaces of $S_k$ and $\Omega_k$, despite adhering to a unified task API.

## B.5. Detailed VQ-VAE Loss and Coverage Regularization

The standard VQ-VAE loss is:

$$\begin{aligned}
\mathcal{L}_{\mathrm{VQ}}(\phi, \boldsymbol{e}) = \\
\|f_\phi^d(x_q) - s\|_2^2 + \lambda_{\mathrm{vq}} \|\mathrm{sg}[x] - x_q\|_2^2 + \lambda_{\mathrm{commit}} \|x - \mathrm{sg}[x_q]\|_2^2,
\end{aligned} \tag{29}$$

where the three terms correspond to reconstruction error, vector quantization alignment, and encoder commitment, respectively. To improve codebook coverage in sparse multi-task settings, we introduce an additional coverage term:

$$\mathcal{L}_{\mathrm{cvr}} = \frac{1}{|\mathcal{J}(t,k)|} \sum_{j \in \mathcal{J}(t,k)} \|\mathrm{sg}[f_\phi^e(s_t^k)] - e_j\|_2^2, \tag{30}$$

*Table 3.* Parameter setting

| Parameter | Value | Parameter | Value |
|---|---|---|---|
| Learning Rate (Actor) | 0.0025 | Learning Rate (Critic) | 0.0005 |
| Discount factor | 0.9 | Soft update factor | 0.001 |
| Optimizer | Adam | Memory length | 50000 |
| Sample size | 32 | Batch size | 32 |

where $\mathcal{J}(t, k)$ denotes a group of embeddings indexed by timestep $t$ and latent task class $k$. $\{e_1, \ldots, e_{n_c}\} \subset \mathbb{R}^d$ denotes the learnable codebook of discrete embeddings. This differs from LAGMA, which groups only by timestep, and follows the motivation of (Na & Moon, 2024; Na et al., 2025) to ensure all codebook entries remain active for robust clustering and cross-task generalization. The final training objective is:

$$\mathcal{L}_{\text{VQ}}^{\text{tot}} = \mathcal{L}_{\text{VQ}} + \lambda_{\text{cvr}}\mathcal{L}_{\text{cvr}} + \lambda_{\text{ent}}\mathcal{L}_{\text{ent}}. \tag{31}$$

The index $k$ is obtained from unsupervised trajectory clustering, and $\text{sg}[\cdot]$ denotes the stop-gradient operator. This task-aware indexing (by $t$ and $k$) differs from LAGMA's timestep-only grouping and promotes broader cross-task codebook utilization (Na & Moon, 2024; Na et al., 2025).

# C. Additional Proposed Methods and Detailed Formulations

## C.1. Pseudocode of TACTIC

# D. Experimental Settings

## D.1. Detailed Experimental Settings

We adopt TRAMA's experimental setup with the following modifications for stability: (1) number of trajectory clusters reduced $8 \to 4$; (2) VQ-VAE codebook size reduced $512 \to 256$; (3) VQ-VAE update interval increased $10 \to 100$; (4) codebook update interval increased $40 \to 100$. Other settings (latent dimensionality, loss weightings, batch size, optimizer) remain consistent. The hyperparameters and comparison algorithms are shown in Tables 3 and 4, respectively. Since TRAMA has already studied codebook size and update frequency in detail, we do not repeat those studies extensively; we only report a compact sensitivity analysis for the key cluster number ncl in Appendix D.4.4.

**Impact of modifications.** Fewer clusters improve efficiency but may slightly reduce task separability. Smaller codebook reduces representational granularity but has minor performance impact. Longer update intervals stabilize embeddings, at the cost of slower convergence.

These adjustments do not materially affect aggregate performance. To ensure fairness, TRAMA's hyperparameters are also adapted, and all ablations are run under the same modified settings.

## D.2. Detailed Ablation Analyses

Interestingly, in the reSurComb3 and reSurComb4 tasks—constructed by reflecting and reordering unit compositions—Static-G surpasses full TACTIC in final returns, while Rand-G exhibits high variance yet occasionally achieves near-optimal performance. This contrasts with the standard SurComb settings, where TACTIC consistently dominates. We attribute this to semantic misalignment in trajectory encodings: the mirrored reSurComb tasks disrupt the class-to-task structure learned by TACTIC's VQ-VAE, weakening graph conditioning. In contrast, Static-G's fixed fully connected topology, though less efficient, maintains stable coordination unaffected by latent drift. Rand-G, despite its instability, introduces stochasticity that may act as exploration regularization, helping avoid overfitting to specific coordination priors. These findings suggest that while trajectory-conditioned graphs are generally effective, their success depends on the stability of the latent abstraction. In tasks with altered semantics or symmetry, simpler or randomized coordination structures may occasionally prove more robust.

*Table 4.* Comparison Algorithms

| Framework | Features | Details |
|---|---|---|
| Subtask-based | RODE (Wang et al., 2021b) | Restricted the exploration space by pre-training classified agents represented as roles. |
| | LIIR (Du et al., 2019) | Designs an individual reward allocation mechanism to prevent the agent training from entering the suboptimal. |
| | TRAMA (Na et al., 2025) | VQ-VAE trajectory codes directly condition both agents and the mixer, enabling task-label-free generalization. |
| Value-Decomposition | QMIX (Rashid et al., 2018) | Uses a hypernetwork-conditioned monotonic mixing network to combine individual utilities into a joint action-value. |
| | QTRAN (Son et al., 2019) | Transforms the joint action-value function into a factorizable form that aligns individual and joint optimal actions. |
| | QPLEX (Wang et al., 2021a) | Introduces a duplex dueling network to factorize the joint action-value function. |
| | GraphMix (Naderializadeh et al., 2020) | Proposes a graph-based value decomposition network built on QMIX. |
| | QATTEN (Yang et al., 2020) | Applies attention mechanisms to learn effective value function factorization. |
| Memory-based | EMC (Zheng et al., 2021) | Utilizes curiosity-driven intrinsic rewards based on prediction errors of individual $Q$-values. |
| | LAGMA (Na & Moon, 2024) | A coverage-regularized VQ-VAE supplies latent-goal intrinsic rewards, driving exploration under sparse rewards. |
| Coordinate Relationships | G2ANet (Liu et al., 2020) | Design hard and soft attention mechanisms to control communication. |
| | LTSCG (Duan et al., 2024) | Constructs sparse temporal coordination graphs by leveraging agents' historical observations. |
| | DICG (Li et al., 2021) | Dynamically infers a coordination graph structure using self-attention. |
| | DMCG (Gupta et al., 2026) | A meta-learned attention graph dynamically captures multi-hop agent dependencies for flexible coordination. |

*Table 5.* Ablation results of TACTIC and its variants on SMACv1 5m_vs_6m and 27m_vs_30m. For each method, we report the mean episode return, its standard deviation (std), estimated steps to convergence (M: million env steps), and return fluctuation range (max-min over the last 500k steps).

| Method | 5m_vs_6m | | | | 27m_vs_30m | | | |
|---|---|---|---|---|---|---|---|---|
| | Return | Std | Steps to Conv. | Fluctuation | Return | Std | Steps to Conv. | Fluctuation |
| FULL (TACTIC) | 14.71 | 0.10 | 2.0M | 1.2 | 14.67 | 1.21 | 2.0M | 2.0 |
| -TA | 12.80 | 1.10 | 3.0M | 2.3 | 12.50 | 1.34 | 3.2M | 2.9 |
| -TCov | 13.50 | 0.85 | 2.7M | 1.6 | 13.70 | 1.22 | 2.6M | 2.3 |
| Static-G | 12.30 | 1.30 | 3.5M | 2.7 | 11.95 | 1.55 | 3.7M | 3.4 |
| Rand-G | 11.90 | 1.35 | 3.7M | 2.8 | 11.30 | 1.60 | 3.9M | 3.6 |
| -TCP | 13.90 | 0.81 | 2.3M | 1.4 | 13.96 | 1.09 | 2.3M | 2.0 |
| -TA-Static | 11.10 | 1.50 | 4.0M | 3.0 | 10.60 | 1.75 | 4.2M | 4.0 |

*Table 6.* Mean episode return ($\pm$ std) on 10gen_terran under different $\lambda$ hyperparameter settings in TACTIC.

| Setting | $\lambda_{cvr}$ | $\lambda_{commit}$ | $\lambda_{vq}$ | $\lambda_{\psi}$ | $\lambda_{smooth}$ | Reward ± Std |
|---|---|---|---|---|---|---|
| Full/Best | 0.25 | 0.25 | 1.0 | 0.1 | 0.1 | $13.85 \pm 0.15$ |
| No Coverage | 0 | 0.25 | 1.0 | 0.1 | 0.1 | $11.80 \pm 1.20$ |
| No Commit | 0.25 | 0 | 1.0 | 0.1 | 0.1 | $12.30 \pm 1.10$ |
| Low VQ | 0.25 | 0.25 | 0.5 | 0.1 | 0.1 | $13.40 \pm 0.30$ |
| High VQ | 0.25 | 0.25 | 2.0 | 0.1 | 0.1 | $13.20 \pm 0.25$ |
| No Psi | 0.25 | 0.25 | 1.0 | 0 | 0.1 | $13.20 \pm 0.22$ |
| Low Psi | 0.25 | 0.25 | 1.0 | 0.05 | 0.1 | $13.65 \pm 0.18$ |
| High Psi | 0.25 | 0.25 | 1.0 | 0.2 | 0.1 | $13.60 \pm 0.25$ |
| No Smooth | 0.25 | 0.25 | 1.0 | 0.1 | 0 | $12.90 \pm 0.95$ |
| High Smooth | 0.25 | 0.25 | 1.0 | 0.1 | 0.2 | $13.60 \pm 0.20$ |
| High Coverage | 0.5 | 0.25 | 1.0 | 0.1 | 0.1 | $13.30 \pm 0.32$ |

---

**Algorithm 1** TACTIC: Task-Aware Sparse Coordination Graphs for Multi-Task Multi-agent Reinforcement Learning

---

1: Initialize encoder $f_\phi^e$, decoder $f_\phi^d$, codebook $\{e_j\}_{j=1}^k$, classifier $f_\psi$, frozen goal predictor $f_{\theta_g}^g$, policy $\pi_i$, Q-networks $Q_\theta^i$, mixer $Q_\theta^{\mathrm{tot}}$, and replay buffer $\mathcal{D}$
2: **for** each episode **do**
3:    **for** each timestep $t$ **do**
4:       **for** each agent $i$ **do**
5:          Encode observation: $x_i^t = f_\phi^e(o_i^t)$
6:          Quantize embedding: $z_i^t = \arg\min_j \|x_i^t - e_j\|_2$
7:          (Optional) Reconstruct: $\hat{o}_i^t = f_\phi^d(e_{z_i^t})$
8:       **end for**
9:    **end for**
10:   Store trajectory $\{o^t, a^t, r^t, o^{t+1}\}_{t=0}^T$ in $\mathcal{D}$
11:   **if** episode mod $C == 0$ **then**
12:     Sample $M$ trajectories $\{\tau^m\}_{m=1}^M$ from $\mathcal{D}$
13:     **for** each trajectory $\tau^m$ **do**
14:       Compute embedding: $\bar{e}^m = \sum_t e_{z_t^m}$
15:     **end for**
16:     Run K-means on $\{\bar{e}^m\}_{m=1}^M$, assign cluster labels $\bar{k}_m$
17:     Assign pseudo-labels $\bar{k}_m$ to $\bar{e}^m$ for next epoch
18:   **end if**
19:   **for** each training step **do**
20:     Compute episode-level embedding $\bar{e} = \sum_{t=0}^T e_{z_t}$
21:     Sample $(o, a, r, o') \sim \mathcal{D}$
22:     Compute $g = f_{\theta_g}^g(o)$                        // frozen, no gradient
23:     Compute $g' = f_{\theta_g}^g(o')$
24:     Compute quantized embeddings $z_i^t = \mathrm{Quantize}(f_\phi^e(o_i^t))$
25:     Predict trajectory class: $\hat{k} = f_\psi(\bar{e}), \quad p(k) = \mathrm{softmax}(\hat{k})$
26:     Generate task-gated graph weights $W_\phi^{(k)}$ and adjacency $\boldsymbol{A}^{(t)} = \sum_{k=1}^{n_{cl}} p(k) \, \mathrm{Softmax}\big(W_\phi^{(k)}[z_i; z_j]\big)$
27:     Construct augmented obs $o_i^{\mathrm{aug}} = [\, o_i; q_i \,]$
28:     Evaluate $Q_\theta^i(o_i^{\mathrm{aug}}, a_i, g_i)$ and mix via QMIX: $Q_\theta^{\mathrm{tot}}(o^{\mathrm{aug}}, a, g)$
29:     Compute target: $y = r + \gamma \max_{a'} Q_{\theta^-}^{\mathrm{tot}}(o', a', g')$
30:     Compute TD loss: $\mathcal{L}_{\mathrm{TD}} = (Q^{\mathrm{tot}} - y)^2$
31:     Compute $\mathcal{L}_{\mathrm{VQ}}^{\mathrm{tot}}$ using reconstruction, vector-quantization, commitment, coverage, and entropy regularization terms.
32:     Compute classifier loss $\mathcal{L}_{\mathrm{cls}} = \mathrm{CrossEntropy}(f_\psi(\bar{e}), \bar{k})$
33:     Compute representation and RL losses:
34:       $\mathcal{L}_{\mathrm{rep}} = \lambda_{\mathrm{VQ}} \mathcal{L}_{\mathrm{VQ}}^{\mathrm{tot}} + \lambda_{\mathrm{cls}} \mathcal{L}_{\mathrm{cls}}$
35:       $\mathcal{L}_{\mathrm{RL}} = \mathcal{L}_{\mathrm{TD}} + \lambda_{\mathrm{smooth}} \mathcal{L}_{\mathrm{smooth}}$
36:     Backpropagate $\mathcal{L}_{\mathrm{rep}}$ to update $\phi, \psi$, and $\mathcal{L}_{\mathrm{RL}}$ to update $\theta$ and the graph module; keep $\theta_g$ frozen.
37:   **end for**
38: **end for**

---

**Limitations in Overlapping and Ambiguous Task Settings.** Similar to the findings reported in TRAMA (Na et al., 2025), we observe that the performance of unsupervised trajectory clustering degrades in tasks with highly overlapping or mirrored unit compositions, such as the reSurComb scenario in SMAC. In these cases, the latent representations of different tasks may overlap in the embedding space, resulting in reduced cluster purity and lower downstream policy performance (see also TRAMA Figure 9, 10 and Sec. 5.2). Our ablation experiments in these settings confirm that while our approach maintains strong generalization in well-separated tasks, additional mechanisms may be required to fully address the limitations in ambiguous environments. Promising directions include the incorporation of partial supervision, adaptive clustering, or auxiliary task-specific signals, as discussed in TRAMA's conclusion and future work.

Ablation results (Table 5) show that removing either trajectory abstraction or dynamic graph topology degrades performance, with the largest drop when both are absent. Structured coordination and semantic grounding are thus jointly essential to TACTIC's effectiveness.

Table 6 shows that TACTIC maintains stable performance across a range of hyperparameter configurations. Removing coverage loss or commitment loss (No Coverage, No Commit) leads to the most noticeable drops in return, confirming their critical roles in latent diversity and regularization. The framework is relatively robust to variations in $\lambda_{\mathrm{vq}}$ and $\lambda_\psi$, with

*Table 7.* Static-graph comparison on mirrored reSurComb variants.

| Method | reSurComb3 Return | reSurComb4 Return |
|---|---|---|
| TACTIC | $2.50 \pm 0.18$ | $2.47 \pm 0.16$ |
| TACTIC + per-episode static graph | $2.69 \pm 0.14$ | $2.51 \pm 0.15$ |
| Static-G | $2.72 \pm 0.13$ | $2.53 \pm 0.14$ |

*Table 8.* Sensitivity to the pruning budget $r$ on SMACv2 SurComb3.

| $r$ | Return |
|---|---|
| 2 | $2.58 \pm 0.12$ |
| 4 | $2.74 \pm 0.13$ |
| 6 | $2.80 \pm 0.11$ |
| 8 | $2.76 \pm 0.14$ |

mild performance peaks observed around moderate values. Similarly, trajectory graph smoothing contributes positively but is not overly sensitive. These results validate the modular stability and generalizability of TACTIC.

**Sensitivity to the pruning budget $r$.** We further evaluate the sensitivity of TACTIC to the sparsity budget $r$ on SMACv2 SurComb3. The pruning budget controls the number of retained coordination edges after variance-based edge scoring. As shown in Table 8, performance remains stable over a moderate range of $r$. A very small budget removes useful coordination edges, while an overly large budget weakens the sparsity advantage. These results support using a fixed sparsity budget across tasks without exhaustive per-task tuning.

### D.3. Ablation Study on Label Consistency and Training Stability

**Sensitivity to K-means initialization.** We first examine whether the trajectory clustering stage is sensitive to different K-means initializations. On SMACv2 SurComb3, we run the clustering procedure with different initialization seeds while keeping the remaining training pipeline unchanged. As shown in Table 9, the final returns vary only slightly across seeds, suggesting that K-means initialization does not dominate subsequent policy learning. This is consistent with our implementation, where previous centroids are reused across clustering updates to reduce label permutation and class switching.

**Training Instability and Early Stage Label Drift.** Similar to the observations in TRAMA (Na et al., 2025), we note that in the early stage of training, both the embedding distribution and cluster centroids are not yet stable. This can lead to frequent label drift, where trajectories are assigned to incorrect clusters due to the random initialization of the neural network and clustering process. If the model is forced to rapidly minimize the distance between embeddings and the current codebook under such noisy assignments, it may inadvertently guide the main policy network toward suboptimal or unstable feature spaces, potentially causing negative transfer or representation collapse. To address this, it is crucial to adopt mechanisms that stabilize cluster assignments and carefully schedule the associated losses during the initial training phase. Detailed mitigation policies and their empirical effects are discussed in the following sections.

**Stabilization Mechanisms for Reliable Training.** To mitigate the risk of unstable cluster assignments and representation drift in the early training phase, we adopt several stabilization policies inspired by TRAMA (Na et al., 2025) and related literature. First, we employ a warm-up stage where the VQ-VAE encoder and codebook are trained independently before introducing the clustering and policy-related losses. This ensures that the embeddings reach a certain level of consistency prior to being used for trajectory clustering and downstream decision making. Second, during the clustering updates, we utilize centroid initialization techniques—re-initializing cluster centroids based on their previous states—to preserve label consistency across clustering updates, following the approach shown effective in TRAMA (see also their Figure 25 and Appendix D.5). Third, we gradually anneal the weights of the clustering and codebook losses: these losses are initially assigned low weights and are increased progressively as training proceeds, to prevent premature overfitting to noisy assignments. Such mechanisms collectively reduce the negative impact of early-stage label noise and promote stable co-adaptation of the embedding space, cluster assignments, and policy learning.

*Table 9.* K-means initialization sensitivity on SMACv2 SurComb3.

| Initialization seed | Return |
|---|---|
| Seed 1 | 2.74 |
| Seed 2 | 2.71 |
| Seed 3 | 2.76 |
| Seed 4 | 2.72 |
| Mean $\pm$ Std | $2.73 \pm 0.02$ |

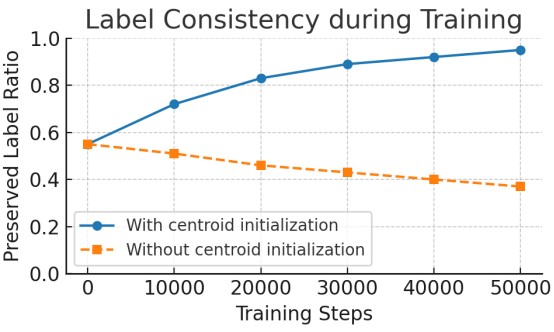 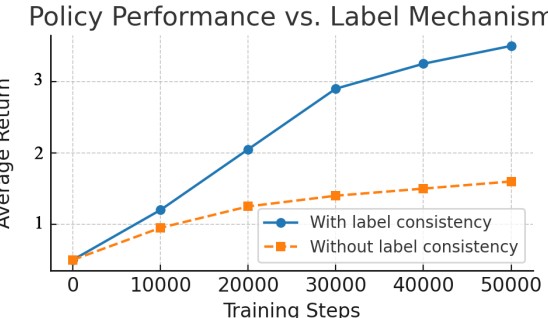

*Figure 5.* (**Left**) Preserved label ratio during training with and without centroid initialization. (**Right**) Policy average return over training steps under different label consistency mechanisms. Our approach with label consistency shows significantly improved cluster stability and policy performance, consistent with observations in TRAMA.

Figure 5 (left) shows the preserved label ratio during training in SurComb4. The use of centroid initialization greatly reduces label drift, leading to more stable clustering. Correspondingly, Figure 5 (right) demonstrates that policy performance is substantially improved when the label consistency mechanism is applied. These findings align with ablation results reported in TRAMA, and empirically validate our design choices for robust representation and policy learning.

### D.4. Extended Experiments

In addition to standard multi-agent benchmarks, we conduct a set of extended evaluations to assess the efficiency, scalability, and robustness of TACTIC under practical deployment scenarios. This section presents runtime profiling, model resource analysis, graph sparsity and abstraction accuracy, as well as generalization under perturbations.

#### D.4.1. RUNTIME EFFICIENCY ANALYSIS

To evaluate computational overhead, we compare TACTIC with representative baselines on per-step latency and graph complexity. Metrics include:

- **Graph Construction Time (ms)**: Average cost to infer the coordination graph.

- **Forward Pass Time (ms)**: Full model inference latency.

- **Active Edges per Step**: Number of communication links.

- **Max-Sum Iterations**: Required message-passing steps for graph resolution.

TACTIC maintains a significantly lower number of active edges and convergence iterations than DMCG and GraphMix, reflecting the benefits of sparse and task-aware graph generation.

#### D.4.2. MODEL COMPLEXITY AND RESOURCE USAGE

We report total model parameter count and peak GPU memory usage during evaluation.

TACTIC achieves high modeling capacity while keeping inference memory footprint lower than DMCG.

*Table 10.* Coordination efficiency across methods (SMACv2 SurComb3).

| Method | Graph Time | Fwd Time | Edges | Max-Sum Iter |
|---|---|---|---|---|
| TACTIC | **1.7 ms** | **3.4 ms** | **6.2** | **2.1** |
| DMCG | 4.8 ms | 5.7 ms | 12.0 | 5.3 |
| GraphMix | 6.1 ms | 6.8 ms | 15.0 | 4.8 |
| TRAMA | — | 3.5 ms | — | — |
| QMIX | — | 2.9 ms | — | — |

*Table 11.* Model complexity and memory usage (SUMO 8 agents).

| Method | Parameters | Peak GPU (MB) |
|---|---|---|
| TACTIC | 0.3M | 507 |
| DMCG | 0.4M | 804 |
| TRAMA | 0.2M | 376 |
| QMIX | 0.1M | 220 |

### D.4.3. COMPUTATIONAL COST AND EFFICIENCY ANALYSIS

*Table 12.* Computational resource and wall-clock time comparison between TACTIC and baselines on SMAC SurComb4.

| Method | Training Time (1M steps, h) | GPU Memory (GB) | CPU Utilization (%) |
|---|---|---|---|
| QMIX | 2.1 | 3.8 | 62 |
| RODE | 2.7 | 5.1 | 68 |
| TRAMA | 3.4 | 6.2 | 70 |
| LAGMA | 3.8 | 7.0 | 73 |
| TACTIC | 4.2 | 7.5 | 74 |

Table 12 summarizes the computational resource usage and wall-clock training time for TACTIC and representative baselines on the SMAC SurComb4 scenario. While TACTIC introduces additional overhead in terms of training time and GPU memory—primarily due to the joint optimization of VQ-VAE, trajectory clustering, and dynamic graph modules—the increase is moderate compared to LAGMA and remains within the range observed for advanced multi-agent reinforcement learning algorithms. Importantly, our implementation decouples the training and inference phases: During inference, the VQ-VAE encoder and trajectory-class predictor are fixed, while the graph module can be used in different update modes: per-step adaptive, low-frequency, or per-episode static. The static mode evaluates only a lightweight policy with a fixed graph, whereas adaptive modes retain limited online graph recomputation for better performance. As a result, TACTIC keeps inference latency close to TRAMA while introducing higher memory usage than QMIX, reflecting a practical trade-off between adaptive coordination and resource cost. These findings are consistent with observations reported in TRAMA, further validating the scalability of our framework.

### D.4.4. SENSITIVITY TO THE NUMBER OF TRAJECTORY CLUSTERS.

Figure 6 presents the results of a sensitivity analysis with respect to the number of trajectory clusters ($n_{cl}$) on SurComb4 and protoss_5_vs_5 tasks. We observe that the average return is highly dependent on the choice of $n_{cl}$. Specifically, TACTIC achieves peak performance when $n_{cl}$ is set to 4 for SurComb4 and 8 for protoss_5_vs_5, which aligns with the inherent diversity and the number of distinct unit compositions in these environments. Choosing a cluster number that is too small or too large leads to a noticeable decline in performance, likely due to over-aggregation or unnecessary splitting of trajectory classes, respectively. These results are consistent with observations reported in TRAMA, and highlight the importance of tuning $n_{cl}$ according to the underlying task structure for optimal generalization and policy effectiveness.

### D.4.5. FREEZING THE GOAL PREDICTOR AND TRAINING STABILITY

As shown in Table 13, freezing the goal predictor during training leads to a higher and more stable average return (3.26 vs. 2.79) across multiple runs. When the goal predictor is not frozen, training becomes less stable, and final performance consistently degrades. This empirical result supports our design choice and aligns with previous findings on gradient interference in modular multi-agent learning frameworks.

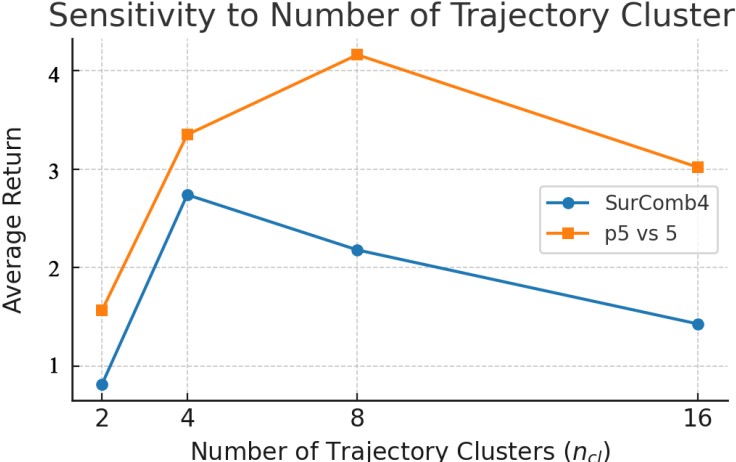

*Figure 6.* Sensitivity analysis of TACTIC to the number of trajectory clusters ($n_{cl}$) on SurComb4 and protoss_5_vs_5 tasks.

*Table 13.* Ablation study on freezing the goal predictor during training. Average return is measured over 5 independent runs.

| Method | Average Return |
|---|---|
| Freeze Goal Predictor (Ours) | $3.26 \pm 0.12$ |
| No Freezing | $2.79 \pm 0.23$ |

### D.5. Practical Inference Procedure and Engineering Deployment

**Practical Inference and Edge Assignment.** During training, TACTIC allows the coordination graph $A^{(t)}$ to vary over time so that the model can learn context- and task-sensitive dependencies. During deployment, we optionally use a simplified inference policy in which the graph is instantiated once at the beginning of an episode (or once per predicted task cluster) and then kept fixed throughout the episode. This does not change the learned payoff decomposition; it is an engineering trade-off that reduces online communication and graph recomputation overhead at the cost of reduced within-episode adaptability. In Appendix D.9, we report an ablation over graph update frequency (per-step / every-$k$-steps / per-episode static) to quantify this trade-off more explicitly.

When encountering new tasks that were not seen during training, the model can reuse the closest matching static graph from the codebook. If significant task distribution shifts occur, a brief adaptation phase—such as re-running the clustering or dependency prediction offline—may partially recover coordination quality, although performance degradation may still be observed. This trade-off between adaptability and communication efficiency is a common feature among graph-based MARL methods (Na et al., 2025; Gupta et al., 2026), and is key to their real-world deployability.

### D.6. Detailed SUMO Scenario Configuration

In this study, we leverage the Simulation of Urban Mobility (SUMO) (Lopez et al., 2018) to evaluate cooperative adaptive vehicle (CAV) interactions under mixed-autonomy traffic conditions, focusing on communication-driven coordination policies. We set up a four-way intersection with four two-lane segments (incoming and outgoing) under a mixed traffic flow of 300 vehicles/hour/lane, including eight controlled CAVs. The goal is for each CAV to navigate the intersection without collisions, optimizing traffic flow and safety. A visualization of the setup is shown in Figure 7. Each CAV follows the following longitudinal motion model:

$$x_i(t+1) = x_i(t) + v_i(t) \cdot 1 + \frac{1}{2} a_i(t) \cdot 1^2$$
$$v_i(t+1) = v_i(t) + a_i(t) \cdot 1, \tag{32}$$

where $x_i(t)$, $v_i(t)$, and $a_i(t)$ represent the position, speed, and acceleration of vehicle $i$ at time $t$, with a decision interval of

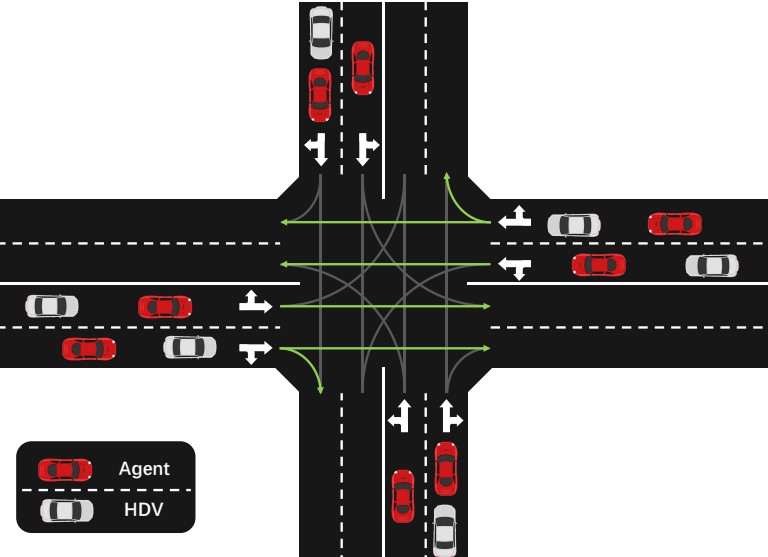

*Figure 7.* Render of SUMO traffic flow.

1s.

To handle CAV coordination, Guo et al. (Guo et al., 2024) assume: a) perfect V2V communication during training with possible delays in testing; b) each CAV's local observation includes position, speed, and binary flags, with partial observability to restrict explicit HDV (Human-Driven Vehicle) data; and c) lateral behavior for CAVs is predefined, with HDVs following fixed routes. Observation space consists of $o_i = \left[x_i, y_i, v_i, d^i_{\min}, t_w, f^i_{\text{enter}}, f^i_{\text{out}}, f^i_{\text{route}}\right]$, where $[x_i, y_i, v_i]$ denotes the CAV's coordinates and speed, $d^i_{\min}$ the minimum distance to the nearest vehicle, $t_w$ the waiting time, and the binary flags $f^i_{\text{enter}}, f^i_{\text{out}}, f^i_{\text{route}}$ track intersection entry, exit, and route choice, respectively. The reward function emphasizes efficiency, safety, and successful completion as follows:

1) Traffic Efficiency ($r_{\text{eff}}$): A penalty for low speed and waiting time, defined as:

$$r_{\text{eff}} = -\sum_{i=1}^{N} \alpha_1 \cdot I\left(v_i < V_{\min}\right) - \sum_{j=1}^{N_l} \alpha_2 \cdot t^j_w - \alpha_3, \tag{33}$$

where $V_{\min}$ represents the minimum acceptable speed, and $\alpha_1, \alpha_2, \alpha_3$ are tunable parameters to balance traffic flow and minimize waiting.

2) Collision Avoidance ($r_{\text{ca}}$): A penalty applied in the event of a collision, calculated as:

$$r_{\text{ca}} = -\sum_{i=1}^{N} \alpha_4 \cdot I(\text{ Collision }). \tag{34}$$

3) Task Completion ($r_{\text{success}}$): A reward for successful intersection passage, based on the remaining time within the episode:

$$r_{\text{success}} = (l_0 - l_{\text{cur}}) \cdot I(\text{ success }) + C + \alpha_5 \cdot n_{\text{pass}}, \tag{35}$$

where $l_0$ is the maximum episode length, $l_{\text{cur}}$ is the current episode length, and $\alpha_5$ is adjustable. The task completes with no collisions and successful intersection crossing. Following the core principles outlined by Guo et al. (Guo et al., 2024),

*Table 14.* Configuration settings of CAV

| Parameters | Grid |
|---|---|
| Lane Length | 100 m |
| Lane Width | 3.2 m |
| Length of CAV | 5 m |
| Maximal Allowed Speed of CAV | 15 m/s |
| Initial Speed of CAVs | 3 m/s |
| Minimal Allowed Speed of CAV | 2 m/s |
| Acceleration Granularity | $[0.3, 2.5, 3.5]$m/s$^2$ |
| Deceleration Granularity | $[-0.3, -2.5, -3.5]$m/s$^2$ |

we design a single randomized-route scenario to evaluate the generalization ability of connected and automated vehicles (CAVs) under dynamic task conditions. In this setting, each CAV randomly selects between a primary and a secondary route within its incoming lane, with a probability vector $[0.7, 0.3]$. To further assess the robustness of the trained policies against communication imperfections, we introduce a 600ms vehicle-to-vehicle (V2V) communication delay during testing, applying a zero-shot transfer evaluation protocol. Table 14 shows the parameters of SUMO (Guo et al., 2024).

4) Environment Rationale

- **Multi-task nature**: Although framed as a single environment, the stochastic arrival times, lane assignments, and right-of-way priorities create a set of implicit sub-tasks, similar to the SurComb suite in TRAMA. Each episode may involve different agent roles (leader, follower, yield, merge), requiring the policy to generalize across diverse interaction patterns.

- **Sparse completion signal**: Although the reward includes efficiency and safety penalties, the main positive completion signal is only obtained when CAVs safely exit the intersection without collisions or deadlock. This sparsity challenges methods relying on dense value shaping or shallow exploration policies.

- **Coordination dependency**: Unlike isolated action settings, intersection navigation requires real-time coordination across multiple agents to avoid gridlocks. Thus, topology-aware methods like TACTIC are particularly well-suited.

- **Realism and adaptive coordination**: The environment models mixed traffic behavior under realistic physical constraints, including HDVs and CAVs. The need to infer sub-goals and communication dependencies reflects adaptive coordination requirements in real-world traffic control.

### D.7. Practical Estimation of Edge-Importance Scores

For each latent task cluster $k$, we estimate the edge-importance score $\zeta_{ij}^{(k)}$ in Eq. (8) from replay/rollout minibatches assigned to cluster $k$. Concretely, for each sampled joint history $\tau_{ij}$, we evaluate the pairwise payoff $q_{ij}(\tau_{ij}, a_i, a_j)$ over the candidate actions of agent $j$ under the current policy $\pi_j$, and compute the empirical variance over $a_j$. We then average these empirical variances across samples in the minibatch to obtain $\hat{\zeta}_{ij}^{(k)}$. To reduce estimation noise, graph updates are performed every $K_g$ environment steps rather than at every optimizer step, and the resulting adjacency is regularized by the temporal smoothing loss in Eq. (14). Unless otherwise stated, the graph retains the top-$r$ edges ranked by $\hat{\zeta}_{ij}^{(k)}$ within each cluster. In our implementation, the replay buffer size for edge-score estimation is 5000, and graph-related updates are performed every $K_g = 5$ environment steps unless otherwise stated.

### D.8. Component-wise Comparison between TRAMA and TACTIC

Since TACTIC shares the trajectory abstraction backbone with TRAMA, we provide a component-wise comparison to clarify the overlap and the technical additions introduced by TACTIC in Table 15.

This controlled comparison shows that the gain does not come from inserting a dynamic graph into a TRAMA-style backbone alone, but from the full semantic-to-structure coupling design.

### D.9. Practical cost of different graph update policies on SMACv2 SurComb3

Table 17 reports the measured cost and reward under different deployment-time graph update policies on SurComb3.

*Table 15.* Component-wise comparison between TRAMA and TACTIC. TRAMA mainly uses latent trajectory classes to condition policy execution, whereas TACTIC further maps latent trajectory semantics to adaptive coordination structure.

| Component | TRAMA | TACTIC |
|---|:---:|:---:|
| VQ-VAE trajectory abstraction | ✓ | ✓ |
| K-means / trajectory clustering | ✓ | ✓ |
| Trajectory-class-aware policy conditioning | ✓ | ✓ |
| Coverage-aware codebook regularization | ✓ | ✓ |
| Trajectory-class predictor | ✓ | ✓ |
| Frozen predictor during policy training | | ✓ |
| Dynamic coordination graph | | ✓ |
| Semantic-conditioned graph modulation | | ✓ |
| Variance-based graph pruning | | ✓ |
| Task-conditioned low-rank payoff factorization | | ✓ |
| Coordination-aware observation augmentation | | ✓ |

*Table 16.* Controlled comparison between a TRAMA-style backbone with dynamic graph insertion and full TACTIC on SMACv2 SurComb3.

| Method | Reward | Forward Time (ms) | Graph Time (ms) | Max-Sum Iter. |
|---|:---:|:---:|:---:|:---:|
| TRAMA-style backbone + dynamic graph only | $2.38 \pm 0.15$ | 3.6 | 2.0 | 2.1 |
| TACTIC (per-step adaptive) | $2.80 \pm 0.12$ | 3.4 | 1.7 | 2.1 |

## E. Limitations and Scope

TACTIC is intended for settings where latent trajectory semantics remain reasonably aligned with coordination structure. Under this condition, trajectory-level abstraction can provide reliable task-conditioned signals for graph adaptation and coordinated decision-making. However, when this semantic-to-structure alignment weakens, the graph adaptation mechanism can become less reliable.

This limitation is reflected in the mirrored/reordered reSurComb variants, where the latent trajectory codes may no longer preserve the coordination-relevant task structure as cleanly as in the standard settings. In such cases, simpler static graphs can be more robust than dynamically adapted graphs. We therefore view these settings as a boundary case of the current framework rather than a contradiction of its intended design.

More broadly, TACTIC introduces an engineering trade-off between adaptivity and deployment cost. During training, the coordination graph can vary over time to capture evolving dependencies, while during deployment a simplified static or low-frequency graph update policy may be preferred to reduce online computation. Accordingly, the method is best viewed as a task-semantic-conditioned coordination framework whose benefits are strongest when latent task abstraction remains stable enough to support reliable structure adaptation.

*Table 17.* Practical cost and reward of different deployment-time graph update policies on SMACv2 SurComb3. The per-step adaptive setting corresponds to the default training-time graph construction in TACTIC. The other rows evaluate lower-frequency or static graph updates using the same trained model at inference time, without changing the training procedure.

| Graph update policy | Reward | Forward Time (ms) | Graph Time (ms) | Max-Sum Iter. |
|---|:---:|:---:|:---:|:---:|
| Per-step adaptive | $2.80 \pm 0.12$ | 3.4 | 1.7 | 2.1 |
| Every 5 steps | $2.73 \pm 0.14$ | 3.2 | 1.2 | 2.1 |
| Every 10 steps | $2.65 \pm 0.15$ | 3.1 | 0.9 | 2.1 |
| Per-episode static | $2.50 \pm 0.18$ | 2.9 | 0.5 | 2.1 |

