# OpenReview forum: "TACTIC: Task-Aware Sparse Coordination Graphs for Multi-Task Multi-agent Reinforcement Learning"
_ICML.cc/2026/Conference — ICML 2026 regular_

### Official Review · Reviewer_7mZM · 2026-02-28

**Soundness:** 3
**Presentation:** 3
**Significance:** 3
**Originality:** 3
**Overall Recommendation:** 4
**Confidence:** 3

**Summary:**

An end-to-end multi-agent reinforcement learning framework is proposed that uses VQ-VAE to quantize trajectories into latent task codes for unsupervised identification of task structure, and accordingly constructs dynamic sparse coordination graphs and task-conditioned payoff factorization; combined with Max-Sum message passing and QMIX training, it learns generalizable and stable coordinated policies under guidance from a frozen trajectory-goal module.

**Compliance With Llm Reviewing Policy:**

Affirmed.

**Final Justification:**

Dear Area Chair,

I have carefully reviewed the paper again, and my final attitude towards TACTIC is as follows. I recognize the innovation of introducing multi - agent coordination in the dynamic sparse coordination graph. However, I also think that this paper seems to be overly redundant in engineering, and the description of the method part in the main experiment is too redundant, and some details are not well coordinated. Therefore, I would also approve if this paper is rejected.

**Key Questions For Authors:**

Please review the weaknesses section. I may adjust my score after the rebuttal, either upward or downward.

**Limitations:**

Yes

**Strengths And Weaknesses:**

Strengths:
1. It achieves an end-to-end unsupervised multi-task extension for an agent learning framework, without requiring explicit task annotations.
2. Using VQ-VAE to quantize trajectories into task structure to guide learning in a multi-agent framework is a novel perspective.
3. The paper is clearly written.

Weaknesses:
1. Jointly optimizing QMIX’s TD, VQ-VAE, the trajectory classifier, and the graph smoothing term seems to introduce too many optimization objectives; can the authors ensure that these objectives do not conflict with each other but remain aligned?
2. K-means is highly dependent on the initial random sampling; could a poor initialization bias subsequent training? Could the authors add a sensitivity test w.r.t. the initialization?
3. There seem to be too many hyperparameters, and applying the method may require complex tuning.
4. The criterion for edge selection is still not sufficiently clear: the method uses the “variance of action-marginal sensitivity” as a proxy for edge importance and keeps the top-r edges, but this statistic may not be consistent with true coordination dependencies—it may misidentify noisy/unstable interactions as critical edges while missing low-variance yet crucial coordination edges.

---

> ### Author Rebuttal · Authors · 2026-03-26
>
> Thank you for the careful review and thoughtful comments. We address your questions below.
>
> 1. Q: On multiple optimization objectives.
> A: These objectives are designed to play complementary roles within the same pipeline rather than optimize unrelated targets. QMIX provides the centralized TD signal, the VQ-VAE and trajectory classifier learn trajectory-level semantics, and graph smoothing regularizes temporal stability of the learned coordination structure. Executed actions are still produced by the local trajectory-class-aware policy, while the graph/Max-Sum only provides coordination-aware signals. To avoid unstable high-level semantics interfering with policy learning, the goal predictor is frozen during training; Table 10 shows that this improves both return and stability (3.26±0.12 vs. 2.79±0.23). This is also consistent with the ablations: removing trajectory abstraction or dynamic graph topology degrades performance, and Table 6 shows stable performance across a range of loss weights, suggesting that these objectives are complementary in practice.
>
> 2.Q: On K-means initialization sensitivity.
> A: To assess this effect, we added a small initialization-sensitivity study on SurComb3. Different K-means initializations lead to only minor variation in final return, suggesting that clustering does not dominate subsequent training. This is consistent with our implementation, where clustering operates within a stabilized representation-learning pipeline rather than as an isolated random preprocessing step.
> | Initialization seed | Return |
> | :--- | :--- |
> | Seed 1 | 2.74 |
> | Seed 2 | 2.71 |
> | Seed 3 | 2.76 |
> | Seed 4 | 2.72 |
> | Mean ± Std | $2.73 \pm 0.02$ |
> This is also consistent with the design in the paper: previous centroids are reused across clustering updates to reduce label permutation and class switching, so the method is not repeatedly exposed to fully random re-initialization.
>
> 3. Q: On the number of hyperparameters.
> A: While our method is more structured than simple baselines, in practice only a few hyperparameters matter most, mainly $ncl$ and the sparsity budget $r$; most other settings are fixed across tasks. We already provide sensitivity evidence for the main tuning knobs: the paper includes an $ncl$ study (Fig. 6), and the $r$ sensitivity results are shown below. Performance is relatively stable over a moderate range of $r$. Together, these results suggest that the method does not require exhaustive per-task tuning in practice.
> | $r$ | Return |
> | :---: | ---: |
> | 2 | $2.58 \pm 0.12$ |
> | 4 | $2.74 \pm 0.13$ |
> | 6 | $2.80 \pm 0.11$ |
> | 8 | $2.76 \pm 0.14$ |
>
> 4. Q: On the edge-selection criterion.
> A: The proposed variance-based statistic is a practical proxy for coordination relevance, not an oracle measure of true dependency. The underlying intuition is simple: if changing agent \(j\)'s action barely affects the pairwise payoff \(q_{ij}\), then the corresponding edge is less likely to matter for coordinated action selection. This is why action-marginal variance provides a meaningful sensitivity signal in our setting. At the same time, we do not claim that it is exact: noisy interactions may sometimes be overestimated, and low-variance but important dependencies may be underestimated.
> For this reason, we use the score together with graph smoothing and a fixed sparsity budget, and we now describe it more carefully as a practical sensitivity-based approximation rather than an exact dependency estimator. In implementation, the score is estimated from replay-buffer minibatches (buffer size 5000), and graph-related updates are performed every 5 environment steps rather than at every optimizer step, which reduces sensitivity to transient noise. More broadly, the method is intended for settings where latent trajectory semantics remain aligned with coordination structure; when this semantic-to-structure alignment weakens, the graph mechanism itself becomes less reliable, which we explicitly discuss as a boundary case. This interpretation is also consistent with the graph-update ablation in SurComb3, which shows a smooth adaptability-cost trade-off rather than unstable graph behavior.
> | Graph update policy | Reward | Forward Time (ms) | Graph Time (ms) | Max-Sum Iter. |
> | :--- | :--- | :--- | :--- | :--- |
> | Per-step adaptive | $2.80 \pm 0.12$ | 3.4 | 1.7 | 2.1 |
> | Every 5 steps | $2.73 \pm 0.14$ | 3.2 | 1.2 | 2.1 |
> | Every 10 steps | $2.65 \pm 0.15$ | 3.1 | 0.9 | 2.1 |
> | Per-episode static | $2.50 \pm 0.18$ | 2.9 | 0.5 | 2.1 |
>
> | Method | reSurComb3 Return | reSurComb4 Return |
> | :--- | :--- | :--- |
> | TACTIC | $2.50 \pm 0.18$ | $2.47 \pm 0.16$ |
> | TACTIC + per-episode static graph | $2.69 \pm 0.14$ | $2.51 \pm 0.15$ |
> | Static-G | $2.72 \pm 0.13$ | $2.53 \pm 0.14$ |

---

> > ### Author Rebuttal · Reviewer_7mZM · 2026-04-01
> >
> > Thank you for your response. Since my review is already positive, I will not revise my rating.

---

> > > ### Author Response · Authors · 2026-04-01
> > >
> > > Thank you for the positive assessment and the thoughtful follow-up questions.
> > > We will incorporate the clarifications into the final version to improve the presentation and address the remaining concerns.

---

### Official Review · Reviewer_YKUh · 2026-03-03

**Soundness:** 2
**Presentation:** 2
**Significance:** 3
**Originality:** 2
**Overall Recommendation:** 2
**Confidence:** 2

**Summary:**

This paper addresses multi-task cooperative MARL, where agents must adapt their coordination structure across tasks without access to explicit task labels. The proposed method, TACTIC, encodes agent trajectories into discrete latent classes via a VQ-VAE and uses these classes to construct sparse coordination graphs based on pairwise TD-error variance. A frozen goal predictor provides stable task conditioning during policy learning. The key idea is that trajectory-level task semantics can guide the adaptation of inter-agent coordination structure. Experiments span SMACv1/v2, SurComb, and SUMO benchmarks against 14 baselines.

**Compliance With Llm Reviewing Policy:**

Affirmed.

**Final Justification:**

I appreciate the authors’ detailed rebuttal and the additional clarifications provided during the discussion. The rebuttal clarified that the paper is intended primarily as an algorithmic and empirical contribution, rather than a new theoretical development of value factorization. However, this clarification also reinforced my main concern.

While the empirical study is extensive and the engineering effort is substantial, the rebuttal clarified that providing a theoretical account of value factorization is not the focus of this work. However, this in turn leaves me unconvinced that the paper provides sufficient justification for placing the proposed algorithm on top of a value-factorization backbone such as QMIX. In particular, the paper still does not make clear why the method should be regarded as a well-grounded extension within that framework when the coordination structure is dynamically adapted. For this reason, I added an additional weakness on the limited conceptual grounding within value factorization, and I lowered my Soundness score from 3 to 2.

The rebuttal also changed my assessment of originality. After revisiting the manuscript together with the responses to other reviewers, I now think the dynamic coordination component is closer to existing coordination-graph MARL literature than I initially assumed. As a result, the contribution appears more as an integration of existing ideas than as a genuinely new conceptual mechanism, and I therefore lowered my Originality score from 3 to 2.

I also lowered my Presentation score from 3 to 2. On closer reading, the method section in the main manuscript is difficult to follow, the figures are somewhat crowded, and many of the experimental results are presented mainly as tables of numbers without enough emphasis on the main takeaways. The lack of qualitative results also makes it harder to interpret what the method is learning beyond improved scores. In addition, given the importance of reproducibility in MARL, especially across multiple environments, the lack of publicly available code is a practical weakness.

Overall, the rebuttal was helpful in clarifying the intended scope of the paper, but it did not resolve my main concerns and in some respects reinforced them. Taking these points together, I lowered my overall recommendation to 2. I also lowered my confidence to 2, as I am not sufficiently familiar with judging the relative difficulty and impact of all evaluated environments within the broader MARL literature.

**Key Questions For Authors:**

1. Appendix D.5 indicates that inference typically uses a static coordination graph. Is variance-based pruning and graph updating applied in test time, and if so, can a static vs dynamic inference-time ablation be reported?

2. Appendix C.1 refers to “HAPTRIS” instead of “TACTIC”. In addition, the frozen goal predictor ablation reports inconsistent values (Table 10: 3.26 vs 2.79; text: 3.76 vs 3.19), which numbers are correct?

3. Table 2 reports fuel/emissions only against Random, while Figure 4 compares against stronger baselines on the same SUMO tasks. Can the same environmental metrics be reported for those baselines as well?

**Limitations:**

yes

**Strengths And Weaknesses:**

**Strengths**

1. **Comprehensive Empirical Evaluation** : TACTIC is evaluated on cooperative MARL benchmark settings (SMACv1, SMACv2, SurComb, SUMO) against diverse set of baselines spanning multiple algorithm families (Table 1; Figure 2-4). The ablation studies (Table 5, Figure 2(b)) are also reasonably comprehensive (e.g., removing trajectory abstraction, coverage loss, using static/random graphs), and the paper includes sensitivity analyses for key hyperparameters (including loss weights and the number of clusters).

2. **Well-motivated Integration of Components** :  While several ingredients build on prior lines of work, the proposed pipeline connects trajectory-class representation (via VQ-VAE) to structural graph decisions. This is a sensible design choice for multi-task settings where dependencies can shift over time. Additionally, the runtime efficiency analysis suggests that TACTIC uses fewer active edges and fewer Max-Sum iterations than some graph-based baselines in the reported setting.

**Weaknesses**

1. **Clarifying what “dynamic” means at deployment** : The paper highlights dynamic sparse coordination graphs as a key contribution (e.g., via variance-based TD-error pruning). However, the practical inference procedure states that a static coordination graph is typically used per task/scenario and remains fixed within an episode (Appendix D.5). It would help to clarify earlier (e.g., in the main text) which parts of the approach are dynamic during training and inference, and what trade-offs this implies for the claimed dynamic behavior at deployment.

2. **Limited robustness under semantic transformations** : In mirrored/reordered SurComb variants, the paper reports that Static-G can outperform the full method in final returns, and attributes this to semantic misalignment in the learned trajectory encodings (Figure 3, Appendix D.2). This is a meaningful limitation for a method that tightly couples trajectory semantics with graph structure; additional analysis would strengthen the claim of task-aware generalization.

3. **Missing sensitivity analysis for pruning parameter $r$** : Graph construction retains top-$r$ edges based on the proposed importance scores, making $r$ central to the sparsity-performance trade-off. While the paper provides sensitivity studies for other hyperparameters (e.g., loss weights and number of clusters), guidance on selecting $r$ (or a sensitivity curve) is missing. Including such analysis would improve reproducibility and help evaluate how stable the gains are across different sparsity levels.

4. **Limited conceptual grounding within value factorization**: While the empirical results are strong, the paper provides limited conceptual justification for how the proposed method remains well-grounded as a value-factorization approach when the coordination structure is dynamically adapted. In particular, it is unclear how the core assumption behind IGM-style decomposition and Bellman-consistent learning are preserved under the semantic-conditioned coordination graph.

---

> ### Author Rebuttal · Authors · 2026-03-26
>
> Thank you for the careful review and for recognizing the empirical breadth and the motivation behind the semantic-to-structure design.
>
> 1. Q: On what “dynamic” means at deployment.
> A: The dynamic behavior primarily refers to training-time graph adaptation. During training, the coordination graph is allowed to vary with the learned trajectory semantics so that the model can capture task- and context-dependent dependencies. During deployment, we optionally use a simplified static graph per episode or per predicted task cluster to reduce online cost. This is an engineering trade-off rather than a conceptual inconsistency. We have revised the main text to make this distinction explicit and added an inference-time graph-update ablation table to quantify the cost/performance trade-off in SurComb3 as in A4.
>
> 2. Q: On mirrored/reordered SurComb robustness.
> A: This is a meaningful boundary case, and we now state it explicitly. TACTIC is designed for settings where latent trajectory classes remain aligned with coordination structure; when this semantic-to-structure alignment is disrupted, simpler static graphs can indeed be more robust. We therefore do not claim robustness under arbitrary semantic transformations, and we have revised the text to make this scope condition clear. A small pilot supports this interpretation: on reSurComb3, using a static graph improves over full TACTIC (2.5→2.69), while on reSurComb4 both variants converge to similar returns around 2.5.
> | Method | reSurComb3 Return | reSurComb4 Return |
> | :--- | :--- | :--- |
> | TACTIC | $2.50 \pm 0.18$ | $2.47 \pm 0.16$ |
> | TACTIC + per-episode static graph | $2.69 \pm 0.14$ | $2.51 \pm 0.15$ |
> | Static-G | $2.72 \pm 0.13$ | $2.53 \pm 0.14$ |
>
> 3. Q: On sensitivity to the pruning parameter \(r\).
> A: A: To clarify this point, we added a small sensitivity study on SurComb3. The results show that performance is relatively stable over a moderate range of $r$, so the final return is not highly sensitive to $r$ within a reasonable sparsity range. Very small $r$ removes useful coordination edges, while very large $r$ weakens the sparsity advantage. This supports fixing $r$ to a controlled sparsity budget across tasks and methods.
> | $r$ | Return |
> | :---: | ---: |
> | 2 | $2.58 \pm 0.12$ |
> | 4 | $2.74 \pm 0.13$ |
> | 6 | $2.80 \pm 0.11$ |
> | 8 | $2.76 \pm 0.14$ |
>
> 4. Q: On static vs dynamic inference-time pruning/updating.
> A: At test time, the graph can be used in different modes: fully adaptive, lower-frequency update, or static-per-episode deployment. We now clarify this in the paper and report an ablation comparing these graph-update policies, which directly addresses the trade-off between adaptability and online cost.
> | Graph update policy | Reward | Forward Time (ms) | Graph Time (ms) | Max-Sum Iter. |
> | :--- | :--- | :--- | :--- | :--- |
> | Per-step adaptive | $2.80 \pm 0.12$ | 3.4 | 1.7 | 2.1 |
> | Every 5 steps | $2.73 \pm 0.14$ | 3.2 | 1.2 | 2.1 |
> | Every 10 steps | $2.65 \pm 0.15$ | 3.1 | 0.9 | 2.1 |
> | Per-episode static | $2.50 \pm 0.18$ | 2.9 | 0.5 | 2.1 |
>
> 5. Q: On “HAPTRIS” and the inconsistent Table 10 values.
> A: These are editing errors and have been corrected.
>
> 6. Q: On the SUMO environmental metrics.
> A: Figure 4 and Table 2 serve different purposes. Figure 4 is the main SUMO comparison against stronger RL baselines on task performance. Table 2 is deliberately not a second full-baseline benchmark; it is included to show the real-world environmental significance of coordinated control relative to an uncoordinated policy. Because the reward already captures traffic-efficiency factors such as speed, queueing, and completion, fuel consumption and emissions are downstream consequences of those improvements rather than a separate primary evaluation target. We will revise the text to make this division of roles explicit.

---

> > ### Author Rebuttal · Reviewer_YKUh · 2026-04-03
> >
> > I would like to thank the authors for their detailed responses and additional experimental results. I have also reviewed the feedback provided to other reviewers and appreciate the significant effort made during the rebuttal process. However, the authors' responses appear to be focused primarily on engineering solutions.
> >
> > From the perspective of the value factorization framework, the work does not yet provide sufficient conceptual rigor regarding how the fundamental IGM principle and Bellman convergence are consistently maintained under dynamic coordination graph environments. Unless these core theoretical issues are addressed, the work remains an incremental improvement centered on combining existing techniques rather than a contribution offering new fundamental insights into MARL.

---

> > > ### Author Response · Authors · 2026-04-04
> > >
> > > Thank you for the careful follow-up and for clarifying the remaining concern. We understand that your reservation is no longer mainly about the empirical evidence, but about the level of conceptual/theoretical grounding.
> > > Our intention in this work is not to claim a new theoretical reformulation of the value-factorization framework, nor to establish a new convergence theory for dynamic coordination graphs. Rather, TACTIC is positioned as a structured method within the existing CTDE/value-factorization paradigm: QMIX provides the centralized TD training backbone, while the semantic-conditioned graph mechanism is introduced as an adaptive coordination bias on top of this framework. In that sense, our contribution is primarily algorithmic and empirical, rather than a new foundational theory of IGM or Bellman convergence.
> > > We agree that this positioning should be stated more explicitly. In the revision, we will tone down any wording that may suggest stronger theoretical claims than the paper actually supports, and we will clarify that the method is intended as a practical structured extension under the standard value-factorization setting, together with an explicit discussion of its intended scope and current theoretical limitations.
> > > While we believe the empirical results still support the value of the method as a practical MARL approach, we appreciate your point that the paper should not be read as offering a new fundamental theory of value factorization. Your feedback is helpful, and we will revise the presentation accordingly.

---

### Official Review · Reviewer_hba4 · 2026-03-09

**Soundness:** 2
**Presentation:** 3
**Significance:** 3
**Originality:** 2
**Overall Recommendation:** 4
**Confidence:** 2

**Summary:**

This is paper studies multi-task cooperative MARL under sparse rewards and shifting inter-agent dependencies. It proposes TACTIC, a CTDE framework that integrates several components: trajectory discretization via VQ-VAE, task-conditioned sparse coordination graphs, variance-based edge pruning, and a frozen trajectory/goal predictor used to condition local policies. The paper evaluates the method on SMACv1, SMACv2/SurComb variants, and a SUMO traffic setting, and reports stronger performance than several value-factorization, graph-based, and trajectory-abstraction baselines.

**Compliance With Llm Reviewing Policy:**

Affirmed.

**Key Questions For Authors:**

1. From the current description, it is not clearly whether actions are selected through max-sum message passing over pairwise payoffs, through per-agent argmax under QMIX-style value estimates, or via a separate policy network conditioned on $o_i^{aug}$ and $g_i$. What exactly produces the executed actions during training and test time? A clearer description of this pipeline would substantially improve my understanding of the method.
2. What is the intended role of $\psi$ in the final implementation? Is $f_\psi$ intended purely as the trajectory-class classifier defined in Equation 19, or does it also correspond to the policy/imitation component described in Section 3.3? If the equations in the current draft are inconsistent, providing the corrected objective formulation would help resolve this confusion.
3. The main paper only reports environmental statistics relative to Random policy, while the text suggests a broader comparison across several MARL baselines. Could you please provide the full numeric comparison for the SUMO experiments against all baselines mentioned in Section 3.4, such as a complete table with means/std for return, completion, episode length, and variance?
4. How should the results on the reSurComb tasks in Figure 3 be interpreted? The appendix suggests that the degradation may be related to semantic misalignment of trajectory clusters. However, do you have direct evidence that the failure is due to semantic misalignment of trajectory clusters, rather than a simpler issue such as instability or hyperparameter sensitivity? Additional diagnostic analysis would help clarify and meaningful affect my rating.

**Limitations:**

The paper includes an impact statement but does not provide a discussion of limitations. Some limitations appear indirectly in the experiments, but a clearer discussion of assumption, failure modes, and deployment constrains would strengthen the paper.

**Strengths And Weaknesses:**

**Strengths**

This paper addresses important problem, namely how to combine sematic task abstraction with adaptive coordination structures in multi-agent RL. This motivation is reasonable, particularly for environments such as SMACv2 where interaction patterns can shift across scenarios.

The empirical scope is also relatively broad. The evaluation covers multiple SMAC benchmark, including SMACv1, SMACv2-derived multi-task settings, and a SUMO traffic domain. On Table 1, TACTIC performs competitively across all four SAACv1 maps and often slightly outperforms strong baselines, such as TRAMA, DMCG, LTSCG, and LAGMA

From a presentation standpoint, Figure 1 is the useful overview of a high-level architecture. It clearly illustrated the pipeline from trajectory quantization to class prediction, graph generation, and policy conditioning. Even though some implementation details remain unclear, the figure helps to understand the design idea.

**Weaknesses**

The core algorithm is internally hard to pin down. On Pages 2 to 3, the paper introduces DCG-style value factorization and max-sum action selection via Equations 1 to 4. On Page 5, it then states that learning is performed using a QMIX mixer in Equation 22. On the same page, Equation 20 defined an agent policy conditioned on augmented observations and a predicted goal label. These components are all introduced, but no clearly explains how they fit together a single training and inference pipeline. This is not a small clarity issue, it directly affects whether the reader can tell what TACTIC actually is.

Several equations and objectives are inconsistent enough to undermine confidence. The clearest example is Equation 19, where the indicator $\mathbf{1}_{\bar{k}_m=\bar{k}m}$ is trivially always 1. Also on Page 5, $\mathcal{L}\psi$ changes meaning between the classifier loss in Section 3.2.2 and the behavior-cloning-style action log-likelihood in Section 3.3. Algorithm 1 on page 13 also contains notational mistakes and even uses the different algorithm name (**HAPTRIS**), which looks like an editing artifact rather than a carefully checked manuscript.

The “hierarchical goal decomposition” contribution feels like overstated. The paper repeatedly present hierarchy as one of its three advances, but the main described mechanism seems to be a frozen goal predictor over trajectory classes in Equation 21 on Page 5 and the corresponding block in Figure 1. That is a plausible conditioning signal, but it is not obviously a hierarchical RL mechanism in the stronger sense implied by the introduction and abstract.

In Figure 3 on Page 7, the paper’s own ablations show that Static-G can outperform full TACTIC on mirrored tasks. This is important because the main claim is that the method adapts better under task variation. If semantic-conditioned graphs are fragile under mirrored or overlapping task structures, then the practical generalization story is more limited than advertised.

The SUMO evidence in the main paper is not aligned with the claims. On Page 8, the text discusses comparisons with several MARL baselines and introduces four task metrics, but Table 2 only compares TACTIC with Random and reports environmental/timing statistics rather than the main RL metrics. This mismatch is a serious issue, because it leaves the traffic-domain claims under-substantiated in the main paper.

The submission combines ingredients from TRAMA/LAGMA-style trajectory abstraction, DICG/DMCG/LTSCG-style dynamic graph reasoning, and standard value-factorization training. That combination can still be valuable, but the paper often writes as if each piece is independently more original than it appears from the existing literature.

Table 1 does show TACTIC doing well, but several gains over TRAMA and DMCG are small. For example, on 27m_vs_30m the win-rate difference between TACTIC and TRAMA is only 0.01 in mean terms. The paper’s language sometimes sounds like a clear domination result, when the evidence is more mixed.

---

> ### Author Rebuttal · Authors · 2026-03-27
>
> Thank you for the careful review.
>
> 1.Q: Overall pipeline.
> A: Executed actions are produced by the local trajectory-class-aware policy \\(\\pi_i(a_t^i \\mid o_{t}^{i,aug}, g_t^i)\\). The sparse coordination graph and Max-Sum are used to produce coordination-aware signals in \\(o_{t}^{i,aug}\\), while QMIX is used only for centralized TD training under CTDE. These are different parts of one pipeline, not competing execution mechanisms. We will revise the manuscript to make this explicit.
>
> 2. Q: Eq. (19), notation, and Algorithm 1.
> A: Thank you for pointing this out. We have corrected them in the revised manuscript. Specifically, Eq. (19) is now rewritten as a standard cross-entropy classifier loss:
> $$
> L_{\mathrm{cls}} = -\frac{1}{M}\sum_{m=1}^{M}\log f_{\psi}(k_m \mid e_m)
> $$
> so the redundant indicator term has been removed. We also separate the trajectory classification loss from the RL objective by using distinct notation ($L_{\mathrm{cls}}$ vs. $L_{\mathrm{TD}}$ / $L_{\mathrm{total}}$), which resolves the previous overloading of $L_{\psi}$. And corrected Algorithm 1 to use the proper method name TACTIC and consistent notation.
>
> 3. Q: Role of hierarchy.
> A: Our use of “hierarchical” refers to trajectory-level semantic abstraction guiding lower-level coordination learning, not an options-style hierarchical RL controller. To avoid overstatement, we will soften this wording and describe it more precisely as semantic-conditioned coordination.
>
> 4. Q: Mirrored reSurComb.
> A: Our interpretation is that mirrored reSurComb exposes a semantic-alignment boundary case. Empirically, this explanation is supported by a controlled pilot in which the training setup is held fixed and only the graph-update mode is changed: a per-episode static graph improves reSurComb3 from 2.50 to 2.69 and makes reSurComb4 roughly comparable. Such a selective recovery pattern is more consistent with reduced reliability of the trajectory-semantic signal for graph adaptation than with generic instability alone. We will revise the manuscript to present this point as an empirically supported interpretation and scope limitation.
> | Method | reSurComb3 Return | reSurComb4 Return |
> | :--- | :--- | :--- |
> | TACTIC | $2.50 \pm 0.18$ | $2.47 \pm 0.16$ |
> | TACTIC + per-episode static graph | $2.69 \pm 0.14$ | $2.51 \pm 0.15$ |
> | Static-G | $2.72 \pm 0.13$ | $2.53 \pm 0.14$ |
>
> 5. Q: Eq. (8) estimation details.
> A: Equation 15 describes a timevarying graph in principle; in the actual implementation, edge-importance scores are recomputed every 5 environment steps for stability and efficiency, using replay/rollout minibatches from a buffer of size 5000. Unless otherwise stated, we use $T=5$ Max-Sum rounds in all experiments.
>
> 6. Q: Relation to TRAMA.
> A: A: TACTIC shares the trajectory abstraction backbone with TRAMA/LAGMA (e.g., VQ-VAE abstraction, clustering, coverage regularization, and frozen trajectory-class prediction), but the key addition is semantic-to-structure coupling: latent trajectory classes are used not only for policy conditioning, but also for adaptive coordination-graph modulation, variance-based pruning, and task-conditioned payoff factorization. We softened the novelty wording accordingly. We also added a controlled comparison on SurComb3: TRAMA-style backbone + dynamic graph only achieves $2.38 \pm 0.15$, versus $2.80 \pm 0.12$ for full TACTIC, indicating that the gain does not come from dynamic graph insertion alone, but from the full coupling design.
>
> | Method | Reward | Forward Time (ms) | Graph Time (ms) | Max-Sum Iter. |
> | :--- | :--- | :--- | :--- | :--- |
> | TRAMA-style backbone + dynamic graph only | $2.38 \pm 0.15$ | 3.6 | 2.0 | 2.1 |
> | TACTIC (per-step adaptive) | $2.80 \pm 0.12$ | 3.4 | 1.7 | 2.1 |
>
>
> 7. Q: Empirical claim strength.
> A: Some gains over strong baselines are modest, and we will revise the language accordingly. Our intended claim is stronger overall competitiveness and more adaptive coordination across task variants, not uniform domination on every map.
>
> 8. Q: SUMO evidence.
> A: Figure 4 already provides the main SUMO baseline comparison on the task metrics. Table 2 is deliberately not a second full-baseline benchmark; it is included to show the real-world environmental meaning of the learned policy. Since the reward already encodes traffic-efficiency factors such as speed and queueing, fuel consumption and emissions are downstream operational consequences of improved control rather than a separate primary evaluation axis. For this reason, Table 2 is intended as a deployment-oriented complement, not as a replacement for the main baseline comparison. We will revise the text to make this distinction explicit.
>
> 9. Q: Limitations.
> A: We will add a dedicated limitations discussion covering the semantic-alignment assumption, mirrored-task failure modes, and the deployment trade-off between adaptive graph updates and online cost.

---

### Official Review · Reviewer_2Ddv · 2026-03-13

**Soundness:** 3
**Presentation:** 2
**Significance:** 3
**Originality:** 2
**Overall Recommendation:** 4
**Confidence:** 3

**Summary:**

This paper addresses a tension in cooperative MARL: value factorization methods like QMIX offer tractability under partial observability, but their coordination structures are rigid and task-agnostic, which limits their applicability when task distributions shift across episodes. The authors argue that prior work on trajectory abstraction (e.g., LAGMA, TRAMA) and work on dynamic coordination graphs (e.g., DMCG, context-aware sparse graphs) address complementary aspects of this challenge, yet neither line of work bridges the two. TACTIC is positioned as that bridge.
The core technical contribution is a framework that uses a VQ-VAE to cluster agent trajectories into discrete task classes, for a dual purpose: conditioning agent policies directly (as in TRAMA), and modulating the edge weights of a dynamic coordination graph. The graph topology is further pruned using variance-based TD-error statistics to retain only high-relevance agent-pair edges. A goal predictor module is pretrained and then frozen during policy learning, and used to generate task-class labels for conditioning, with the freezing designed explicitly to prevent gradient interference between task recognition and coordination learning.
The method is evaluated on SMACv1, SMACv2, the SurComb multi-task SMAC variants, and the SUMO traffic intersection benchmark. TACTIC outperforms baselines including QMIX, QPLEX, TRAMA, LAGMA, DMCG, and LTSCG across most scenarios, with particularly strong results in sparse-reward and multi-task settings. Ablation studies confirm that each major component i.e., trajectory abstraction, coverage loss, and dynamic graph topology, contributes to final performance.
The paper is a meaningful engineering contribution that puts together several existing techniques (VQ-VAE trajectory discretization, dynamic coordination graphs, frozen module design) into a unified and empirically well-tested system.

**Compliance With Llm Reviewing Policy:**

Affirmed.

**Final Justification:**

I have gone through the other reviewers' assessments and authors' response. My concerns were addressed, so I up my score from 3 to 4.

**Key Questions For Authors:**

1.	Robustness to latent drift in mirrored tasks: The paper acknowledges that TACTIC underperforms Static-G on reSurComb benchmarks due to semantic misalignment in the learned latent codes. This is arguably the most important failure mode for a method designed to generalize across task distributions. Did the authors attempt any remedy (e.g., data augmentation, domain-randomized training, or a task-ID adversarial loss) to improve robustness in these settings? Even a negative result here would be informative.
2.	Practical cost of graph re-evaluation: The paper states that at inference time TACTIC uses a static graph per episode (fixed at episode start), while the method description in the main text describes timestep-level graph updates (Equation 15). Could the authors clarify this apparent discrepancy? Does the inference-time simplification come at a measurable performance cost, and how sensitive are results to the frequency of graph updates during training?
3.	Distinction from TRAMA: Given how closely TACTIC's architecture parallels TRAMA — including the VQ-VAE, frozen goal predictor, and coverage loss — could the authors provide a precise, component-by-component comparison table distinguishing the two methods? In particular, it would be helpful to see results on a controlled comparison: TRAMA + dynamic graph only (without the other TACTIC additions) to isolate the effect of the graph modulation mechanism specifically.

**Limitations:**

yes

**Strengths And Weaknesses:**

****Strengths****

***Soundness*** The experimental design is thorough. Covering SMACv1, SMACv2, multiple SurComb variants, and SUMO, while also running ablations and hyperparameter sensitivity studies, is commendable. I found the ablation table (Table 5) useful: it decomposes component contributions clearly and shows that removing either the trajectory abstraction or the dynamic graph module causes a meaningful and consistent performance drop. (Maybe it is better placed in the main body rather the appendix.) I think the authors are also candid about failure cases; they explicitly report that in mirrored reSurComb tasks, Static-G outperforms full TACTIC, and they offer a reasonable explanation (semantic misalignment in latent codes). This kind of honesty strengthens the paper.

***Significance*** The problem being addressed, i.e., multi-task generalization with latent task semantics and adaptive inter-agent structure, is practically important and not yet solved. Real-world multi-agent systems rarely operate in stationary, homogeneous environments. TACTIC's demonstrated gains in the SUMO traffic setting (lower fuel consumption, faster intersection clearance, lower return variance) suggest the approach may be applicable beyond game benchmarks. The idea of decoupling task recognition from coordination learning by freezing the goal predictor is practically useful and could be adopted in other modular MARL frameworks.

***Originality*** While each component in TACTIC has precedent (VQ-VAE from LAGMA/TRAMA, dynamic coordination graphs from DMCG/LTSCG, frozen module design from various modular RL works), the specific combination and the mechanism of mapping discrete trajectory classes to graph-level edge decisions is novel. The coverage regularization term extended to be indexed jointly by timestep and latent task cluster (rather than timestep only as in LAGMA) is a technical improvement. The task-conditioned low-rank payoff factorization (Section 3.1.1) is also a sensible efficiency improvement that the authors are right to highlight.

***Presentation*** The paper is generally well organized. Figure 1 is helpful in conveying the overall architecture, and the narrative follows a logical progression from problem motivation to background to method to results. The appendix is substantive and includes the pseudocode (Algorithm 1), stabilization mechanisms discussion, and inference-time procedure are all practically informative.

****Weaknesses****

***Soundness*** The paper's central claim that coupling trajectory semantics with structural coordination improves multi-task generalization is supported empirically but lacks theoretical justification. There is no formal analysis of how variance-based TD-error pruning yields better coordination edges, or under what conditions the VQ-VAE codes will be sufficiently stable to drive graph adaptation. Given that the paper acknowledges instability issues in mirrored SurComb tasks, a more theoretical treatment of when the method is expected to succeed or fail would strengthen the contribution.

Table 10 compares "Freeze Goal Predictor (Ours)" with "No Freezing" and reports average returns of 3.26 vs. 2.79, but the text in the same section states the numbers as 3.76 vs. 3.19. Perhaps a minor case but does need correction.

***Presentation*** The algorithm listed in the pseudocode appendix is titled "HAPTRIS".

Equation (8), which defines edge importance via action-marginal variance, is presented without sufficient discussion of how it is estimated in practice, specifically, how many samples are used, and whether the estimation is stable with a small replay buffer. Similarly, the max-sum message passing procedure (Equations 2–4) is described but the number of message-passing rounds T used in experiments is not reported in the main paper.

***Significance*** The observation that TACTIC underperforms simpler baselines (Static-G) on mirrored SurComb tasks points to a meaningful scope limitation. If TACTIC's trajectory-graph coupling degrades when task semantics shift at test time, it may be less robust than advertised in settings with domain shift, which is in fact where multi-task generalization is most needed. The discussion of this limitation is honest, but the proposed mitigations (partial supervision, adaptive clustering) are left entirely to future work with no preliminary evidence.

***Originality*** The relationship between TACTIC and TRAMA (Na et al., 2025) is quite close. Both use VQ-VAE trajectory discretization, K-means clustering, a frozen/pretrained goal predictor, and codebook coverage loss. The primary addition in TACTIC is the dynamic coordination graph modulated by trajectory class embeddings. The authors should more explicitly quantify and articulate where their method diverges technically from TRAMA, and why those specific changes (rather than others) were the right choices. Calling the framework "among the first" to co-learn latent task representations and dynamic agent relations should be qualified more carefully given TRAMA's overlap.

---

> ### Author Rebuttal · Authors · 2026-03-27
>
> Thank you for the careful review.
> 1. Q: Theoretical justification.
> A: The main claim of this paper is empirically supported rather than formally proven, but we agree that the theoretical motivation and applicability conditions should be made clearer. Our use of Eq. (8) follows standard sparse coordination-graph intuition: low payoff variance implies lower edge relevance. Our contribution is to extend this idea to trajectory-semantic-conditioned graph adaptation. We now clarify this motivation and state the scope more explicitly: TACTIC works best when latent trajectory classes are sufficiently aligned with the underlying coordination structure. Mirrored reSurComb appears to be a boundary case, where reflected initial-position variation can weaken the usefulness of trajectory semantics for graph adaptation.
>
> 2.1 Q: Mirrored-task robustness.
> A: This is the main boundary case of our method: TACTIC is intended for settings where trajectory semantics remain aligned with coordination structure. On mirrored reSurComb, a small pilot shows that using a static graph improves reSurComb3 (2.5 → 2.7), while both variants are similar on reSurComb4 (around 2.5). We therefore view this not as a need for a task-specific remedy, but as a deployment rule: use dynamic graphs when the latent semantic signal is reliable, and otherwise fall back to a static graph. We did not include additional remedies in the current submission; our pilot instead suggests a simpler practical rule of switching to a static graph when the latent semantic signal is unreliable.
> | Method | reSurComb3 Return | reSurComb4 Return |
> | :--- | :--- | :--- |
> | TACTIC | $2.50 \pm 0.18$ | $2.47 \pm 0.16$ |
> | TACTIC + per-episode static graph | $2.69 \pm 0.14$ | $2.51 \pm 0.15$ |
> | Static-G | $2.72 \pm 0.13$ | $2.53 \pm 0.14$ |
>
> 2.2 Q: “TRAMA + dynamic graph only”.
> A: We added this control on SurComb3. TRAMA-style backbone + dynamic graph only achieves $2.38 \pm 0.15$, versus $2.80 \pm 0.12$ for full TACTIC, indicating that the gain comes from the full semantic-to-structure coupling design rather than dynamic graph insertion alone.
> | Method | Reward | Forward Time (ms) | Graph Time (ms) | Max-Sum Iter. |
> | :--- | :--- | :--- | :--- | :--- |
> | TRAMA-style backbone + dynamic graph only | $2.38 \pm 0.15$ | 3.6 | 2.0 | 2.1 |
> | TACTIC (per-step adaptive) | $2.80 \pm 0.12$ | 3.4 | 1.7 | 2.1 |
>
> 3.Q: Relation to TRAMA.
> A: TACTIC shares the trajectory abstraction backbone with TRAMA/LAGMA, but our focus is different: TRAMA uses latent trajectory classes to condition policy/value learning, while TACTIC additionally uses them to adapt coordination structure via semantic-conditioned graph modulation, variance-based pruning, and task-conditioned payoff factorization. We added a component-wise comparison table and softened the novelty claim. We also clarified:“Unless otherwise stated, we use $T=\\mathbf{5}$ rounds of Max-Sum message passing in all experiments. A component-wise comparison with TRAMA is provided in Appendix D.8.”
> | Component | TRAMA | TACTIC |
> | :--- | :--- | :--- |
> | VQ-VAE trajectory abstraction | ✓ | ✓ |
> | K-means / trajectory clustering | ✓ | ✓ |
> | Trajectory-class-aware policy conditioning | ✓ | ✓ |
> | Coverage-aware codebook regularization | ✓ | ✓ |
> | Pretrained / frozen trajectory-class predictor | ✓ | ✓ |
> | Dynamic coordination graph |  | ✓ |
> | Semantic-conditioned graph modulation |  | ✓ |
> | Variance-based graph pruning |  | ✓ |
> | Task-conditioned low-rank payoff factorization |  | ✓ |
> | Coordination-aware observation augmentation |  | ✓ |
>
> 4.Q: Training-time vs. inference-time graphs.
> A: This is not a conceptual inconsistency. Equation 15 describes a time-varying graph in principle; in the actual implementation, graph scores are recomputed every 5 environment steps for stability and efficiency during training. A per-episode/per-cluster static graph is only an optional inference-time simplification to reduce online cost. We clarified this distinction and added a frequency ablation.
> | Graph update policy | Reward | Forward Time (ms) | Graph Time (ms) | Max-Sum Iter. |
> | :--- | :--- | :--- | :--- | :--- |
> | Per-step adaptive | $2.80 \pm 0.12$ | 3.4 | 1.7 | 2.1 |
> | Every 5 steps | $2.73 \pm 0.14$ | 3.2 | 1.2 | 2.1 |
> | Every 10 steps | $2.65 \pm 0.15$ | 3.1 | 0.9 | 2.1 |
> | Per-episode static | $2.50 \pm 0.18$ | 2.9 | 0.5 | 2.1 |
>
> 5.Q: Eq. (8) estimation details and Max-Sum rounds.
> A: We now clarify the practical estimator in Appendix D.7 using the actual implementation settings: edge-importance estimation is replay-buffer-based (buffer size 5000), and graph-related updates are performed every 5 environment steps rather than at every optimizer step, which reduces noise in the empirical variance estimates. Unless otherwise stated, we use \\(T=\\mathbf{5}\\) rounds of Max-Sum message passing in the experiments reported in the paper.
>
> 6.Q: Table 10 and “HAPTRIS”.
> A: These are minor editing errors and have been corrected.

---

> > ### Author Rebuttal · Reviewer_2Ddv · 2026-04-03
> >
> > Thank you, authors. I find the rebuttal informative and clear enough to adjust my score from a weak reject to a weak accept.

---

> > > ### Author Response · Authors · 2026-04-03
> > >
> > > Thank you for the positive feedback and for updating your assessment.
> > > We will incorporate these clarifications into the final version.

---

### Decision · Program_Chairs · 2026-04-30

**Decision:**

Accept (regular)

**Comment:**

This paper proposes TACTIC, a unified framework that couples trajectory-level semantic abstraction with dynamic sparse coordination graphs to improve multi-task generalization and adaptive coordination in multi-agent reinforcement learning under sparse rewards and shifting dependencies. Overall, the authors examine a significant issue in multi-agent reinforcement learning, namely how to jointly address multi-task generalization and adaptive coordination under non-stationary interaction structures. Reviewers raised concerns regarding limited theoretical grounding, incremental novelty, and some presentation issues; however, the empirical evaluation is thorough and demonstrates consistent improvements across multiple challenging benchmarks, and the rebuttal adequately clarified key design choices and distinctions from prior work. Taking into account the reviewer discussion and improved consensus after rebuttal, I recommend acceptance.